# The decoy SNARE Tomosyn sets tonic versus phasic release properties and is required for homeostatic synaptic plasticity

Chad W Sauvola[1], Yulia Akbergenova[1], Karen L Cunningham[2], Nicole A Aponte-Santiago[2], J Troy Littleton[1,2]*

[1]Department of Brain and Cognitive Sciences, The Picower Institute of Learning and Memory, Massachusetts Institute of Technology, Cambridge, United States; [2]Department of Biology, Massachusetts Institute of Technology, Cambridge, United States

**Abstract** Synaptic vesicle (SV) release probability ($Pr$) is a key presynaptic determinant of synaptic strength established by cell-intrinsic properties and further refined by plasticity. To characterize mechanisms that generate $Pr$ heterogeneity between distinct neuronal populations, we examined glutamatergic tonic (Ib) and phasic (Is) motoneurons in *Drosophila* with stereotyped differences in $Pr$ and synaptic plasticity. We found the decoy soluble N-ethylmaleimide sensitive factor attachment protein receptor (SNARE) Tomosyn is differentially expressed between these motoneuron subclasses and contributes to intrinsic differences in their synaptic output. Tomosyn expression enables tonic release in Ib motoneurons by reducing SNARE complex formation and suppressing $Pr$ to generate decreased levels of SV fusion and enhanced resistance to synaptic fatigue. In contrast, phasic release dominates when Tomosyn expression is low, enabling high intrinsic $Pr$ at Is terminals at the expense of sustained release and robust presynaptic potentiation. In addition, loss of Tomosyn disrupts the ability of tonic synapses to undergo presynaptic homeostatic potentiation.

*For correspondence:
troy@mit.edu

Competing interest: The authors declare that no competing interests exist.

## Editor's evaluation

Sauvola and colleagues define the function of Tomosyn in establishing release probability at NMJ synapses in *Drosophila* larval NMJs. They present compelling evidence that loss of Tomosyn results in increased evoked and spontaneous release. They further find that Tomosyn likely acts as a decoy SNARE protein independently of Syt 1 and Syt 7 to negatively regulate SV docking. The data are of no doubt interesting for researchers in the synaptic transmission field.

## Introduction

$Ca^{2+}$-dependent fusion of synaptic vesicles (SVs) is the primary mechanism for neurotransmission and is mediated by the soluble N-ethylmaleimide sensitive factor attachment protein receptor (SNARE) family (*Jahn and Scheller, 2006*; *Söllner et al., 1993*; *Sudhof, 2004*; *Weber et al., 1998*). Following an action potential, SNARE proteins located on the SV and plasma membrane zipper into an energetically favorable coiled-coil bundle to induce SV fusion (*Jahn and Scheller, 2006*; *Söllner et al., 1993*; *Südhof and Rothman, 2009*). Neurotransmitter release results in a postsynaptic response that varies in size depending on the strength of the synapse, which can be regulated from both pre-and postsynaptic compartments. The postsynaptic cell controls sensitivity to neurotransmitters by governing

**eLife digest** Nerve cells transmit messages in the form of electrical and chemical signals. Electrical impulses travel along a neuron to the junction between two neighbouring cells, the synapse. There, chemical messengers called neurotransmitters are released from one cell and detected by the next, which can either excite or inhibit the recipient cell. Synapses differ in their ability to propagate signals and their signalling activity also fluctuates at times. Moreover, synaptic connections can be strengthened or weakened in a process called plasticity, which is a key part of learning new skills and recovering from a brain injury. It is thought that synaptic signalling might be amped up or dialled down to change the output of the connection between two cells, but exactly how this happens remains unclear. To investigate why synapses differ and how their signalling capabilities change, Sauvola et al. examined the connections between neurons and muscle cells in developing fruit flies. In fruit fly larvae, two types of neurons – called tonic Ib and phasic Is neurons – form synapses with muscle cells. But their synapses have different signalling properties: Ib synapses are weaker than Is synapses. Sauvola et al. hypothesised that a protein called Tomosyn – which is thought to restrict chemical signalling at the synapse – might be more active at weaker Ib synapses. Sauvola et al. found that Tomosyn was indeed more abundant at Ib synapses than at Is synapses, appearing to reflect their differences in signalling properties. In flies engineered to lack the Tomosyn protein, Ib synapses became four times stronger than usual, while Is synapses hardly changed. This supports the idea that Tomosyn restricts the release of neurotransmitters at typically weak Ib synapses. Further experiments showed Ib synapses in flies lacking Tomosyn also lost their malleability and ability to become strengthened during synaptic plasticity. Though the precise molecular interactions need further investigation, the findings suggest that Tomosyn is required for some forms of synaptic plasticity by controlling how much chemical signal neurons release. In summary, this work advances our understanding of synaptic signalling and brain plasticity, showing once again how the brain can change itself.

receptor field composition, while the presynaptic neuron establishes the probability ($P_r$) of SV fusion (*Citri and Malenka, 2008*; *Körber and Kuner, 2016*; *Bliss et al., 2003*; *Yang and Calakos, 2013*). Highly stereotyped differences in $P_r$ exist across neurons, with many neuronal populations broadly classified as tonic or phasic depending on their spiking patterns, $P_r$ and short-term plasticity characteristics (*Atwood and Karunanithi, 2002*; *Dittman et al., 2000*; *Lnenicka and Keshishian, 2000*). How cell-intrinsic properties establish differences in presynaptic $P_r$ between neuronal classes, and how release strength is further refined via plasticity, remain incompletely understood.

The *Drosophila melanogaster* larval neuromuscular junction (NMJ) provides a robust genetic system for characterizing mechanisms mediating synaptic communication and tonic versus phasic release properties (*Aponte-Santiago et al., 2020*; *Aponte-Santiago and Littleton, 2020*; *Genç and Davis, 2019*; *Lu et al., 2016*; *Newman et al., 2017*; *Wang et al., 2021*). Larval body wall muscles are co-innervated by two glutamatergic motoneuron populations that drive locomotion, including the tonic-like Ib and phasic-like Is subtypes (*Aponte-Santiago et al., 2020*; *Harris and Littleton, 2015*; *Jan and Jan, 1976*; *Lnenicka and Keshishian, 2000*). Tonic Ib terminals display lower initial $P_r$ and sustained release during stimulation, whereas phasic Is terminals show higher intrinsic $P_r$ and rapid depression (*Lu et al., 2016*; *Newman et al., 2017*). The *Drosophila* NMJ also undergoes robust presynaptic homeostatic potentiation (PHP) that rapidly increases $P_r$ to compensate for disruptions to postsynaptic glutamate receptor (GluR) function (*Böhme et al., 2019*; *Gratz et al., 2019*; *Li et al., 2018*; *Müller et al., 2012*; *Ortega et al., 2018*; *Weyhersmüller et al., 2011*). In addition to intrinsic release differences, the Ib and Is subtypes display distinct capacity for PHP (*Newman et al., 2017*; *Genç and Davis, 2019*). How tonic and phasic neurons differentially regulate $P_r$ during normal synaptic communication and plasticity is largely unknown.

The highly conserved SNARE regulatory protein Tomosyn negatively controls SV release and has been proposed to participate in synaptic plasticity (*Ben-Simon et al., 2015*; *Chen et al., 2011*; *Gracheva et al., 2006*; *McEwen et al., 2006*). Tomosyn has an N-terminal WD40 repeat domain and a C-terminal SNARE motif with homology to the SV v-SNARE Synaptobrevin 2 (Syb2) (*Fasshauer et al., 1998*; *Hatsuzawa et al., 2003*; *Hattendorf et al., 2007*; *Pobbati et al., 2004*; *Williams et al., 2011*). Tomosyn inhibits presynaptic release by binding the t-SNAREs Syntaxin1 (Syx1) and SNAP-25

to prevent Syb2 incorporation into the SNARE complex fusion machinery (*Hatsuzawa et al., 2003*; *Lehman et al., 1999*; *Sakisaka et al., 2008*; *Williams et al., 2011*).

To further examine the role of Tomosyn in synaptic transmission and plasticity, we used CRISPR to generate mutations in the sole *Drosophila tomosyn* gene. Structure-function analysis revealed the SNARE domain is critical for release inhibition, while the scaffold region promotes enrichment of Tomosyn to SV-rich sites. Despite enhanced evoked release, *tomosyn* mutants fail to maintain high levels of SV output during sustained stimulation due to rapid depletion of the immediately releasable SV pool. Tomosyn is highly enriched at Ib synapses and generates tonic neurotransmission properties characterized by low $P_r$ and sustained release in this population of motoneurons. Indeed, optogenetic stimulation and optical quantal analysis demonstrate an exclusive role for Tomosyn in regulating intrinsic release strength in tonic motoneurons. PHP expression primarily occurs at tonic synapses and is abolished in *tomosyn* mutants, suggesting Tomosyn is also essential for acute PHP expression. Together, these data indicate Tomosyn mediates the tonic properties of Ib motoneurons by suppressing $P_r$ to slow the rate of SV usage, while decreasing Tomosyn suppression enables $P_r$ enhancement during PHP. Conversely, the absence of Tomosyn in Is motoneurons facilitates phasic release properties by enabling an intrinsically high $P_r$ that quickly depletes the releasable SV pool, resulting in rapid synaptic depression and reduced capacity for PHP.

## Results

### *Drosophila* encodes a single conserved Tomosyn gene with two splice variants

The *Drosophila* Tomosyn homolog is highly conserved with other Tomosyn proteins, displaying high sequence conservation in critical domains including the C-terminal SNARE motif. This region enables the formation of a SNARE complex composed of the Tomosyn C-terminus and the t-SNAREs Syx1A and SNAP-25 (*Fasshauer et al., 1998*; *Hatsuzawa et al., 2003*; *Pobbati et al., 2004*; *Williams et al., 2011*). BLOSUM62 alignment using the C-terminal tail of the yeast homolog Sro7 as an outgroup indicates the Tomosyn SNARE motif forms a phylogenetically distinct group from other v-SNAREs despite their shared affinity for t-SNAREs (*Figure 1A*). Homology modeling suggests *Drosophila* Tomosyn forms a SNARE complex that is similar in structure to mammalian Tomosyn (*Figure 1B*). A conserved ERG sequence within the C-terminal SNARE motif enables zippering of Tomosyn with t-SNAREs in a complex that prevents association with the SV fusion clamp Complexin, in contrast to SNARE complexes containing Syb2 (*Figure 1C and D*).

Similar to other species, *Drosophila tomosyn* is alternatively spliced at exon 13 to generate two splice variants, Tomosyn13A and Tomosyn13B, that encode distinct regions of the WD40 repeat scaffold. The sequence of exon 13A is highly conserved across insect genomes, while the 13B exon sequence is poorly conserved (*Figure 1E*). As such, Tomosyn13A is likely the more functionally conserved isoform. Iterative homology modeling of Tomosyn13A suggests it forms a double-barrel structure with three disordered loops projecting from the core WD40 scaffold domain (*Figure 1F*), as predicted for mammalian Tomosyn-1 and Tomosyn-2 proteins (*Williams et al., 2011*). Exon 13 encodes one of the loops protruding from the WD40 core, indicating alternative splicing regulates secondary features of Tomosyn beyond its SNARE-binding properties. Together, these data suggest *Drosophila* Tomosyn shares conserved features with its mammalian counterparts.

### *Tomosyn* mutants display increased evoked and spontaneous neurotransmitter release

To assay the function of Tomosyn at *Drosophila* synapses, CRISPR/Cas9 was used to generate two mutant alleles of the *tomosyn* gene on the X-chromosome (*Figure 1E*). A deletion mutant of *tomosyn* was generated using homology-directed repair to replace the entire coding sequence with a DsRed cassette (*tomosyn^NA1*). A frame shift mutant with an early stop codon (*tomosyn^FS1*) was also isolated using a gRNA that targets the 5' end of the *tomosyn* coding region. *Tomosyn^NA1* mutants were primarily used in this study, though both alleles displayed similar phenotypes (*Figure 2* and *Figure 2—figure supplement 1*). *Tomosyn^NA1* males are homozygous viable and eclose as adults at similar rates to a genetic background control (n≥95 eclosed flies; Chi-square test, p=0.9163). Homozygous adult females eclosed less frequently, suggesting the existence of sex-divergent roles. Tomosyn

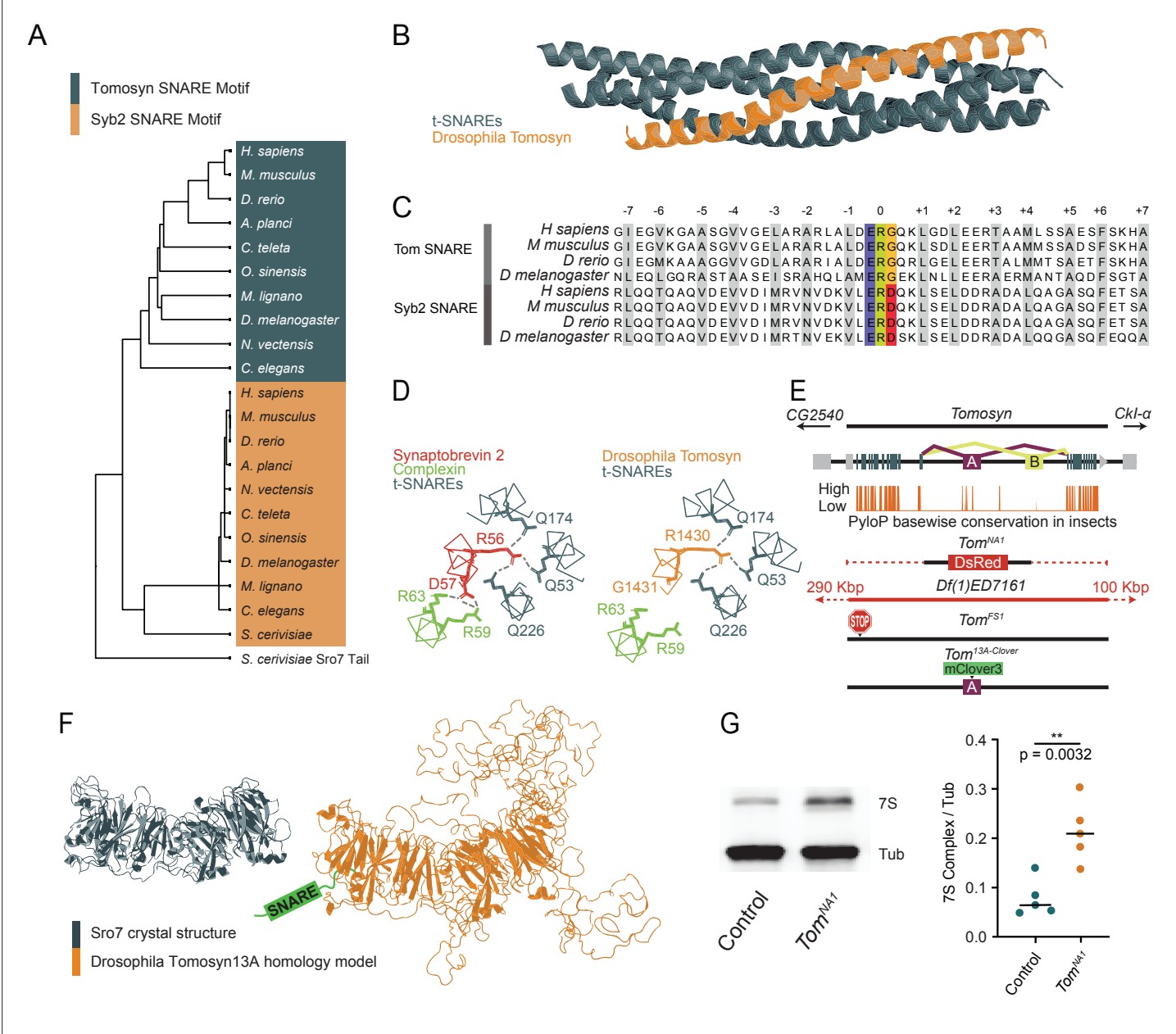

**Figure 1.** Generation of mutations in the conserved *Drosophila* omosyn homolog. (**A**) BLOSUM62 alignment tree of Tomosyn and Syb2 SNARE motifs across the indicated species. The C-terminal tail of *Saccharomyces cerevisiae* Sro7 was used as an outgroup (**B**) Homology model of the *Drosophila* Tomosyn SNARE domain in complex with Syx1A and SNAP-25. (**C**) Sequence alignment between the SNARE domains of Tomosyn and Syb2 from humans (*Homo sapiens*), mouse (*Mus musculus*), zebrafish (*Danio rerio*), and *Drosophila* (*Drosophila melanogaster*). (**D**) Complexin interaction site with the Syb2 SNARE complex compared to the Tomosyn/t-SNARE complex (adapted from data shown in **Figure 5A** of **Pobbati et al., 2004**). (**E**) Genomic structure of *Drosophila tomosyn* shows mutually exclusive splicing at coding exon 13 (top). Basewise conservation of *tomosyn* across insect genomes using PhyloP (middle). Diagram of *tomosyn* CRIPSR mutants, including *tomosyn^{NA1}* that replaces the locus with a DsRed cassette and *tomosyn^{FS1}* with an early frameshift stop codon. A deficiency (*Df(1)ED7161*) spanning the locus is also shown. (**F**) Structure of the *S. cerevisiae* L(2)GL scaffold protein Sro7 (left, adapted from **Figure 1C** of **Hattendorf et al., 2007**) and iterative homology model of *Drosophila* Tomosyn13A (right, adapted from **Figure 1B** of **Williams et al., 2011**). (**G**) Representative Western blot of adult brain lysates stained with anti-Syx1A to label the 7 S SNARE complex and anti-Tubulin as a loading control. The ratio of 7 S complex/Tubulin intensity for control (0.06528, 0.07891±0.01658, n=5 samples, 10 brains per sample) and *tomosyn^{NA1}* (0.2082, 0.2127±0.06183, n=5 samples, 10 brains per sample; Student's t test, p=0.0032) is shown on the right (*Figure 1—source data 1*, *Figure 1—source data 2*, *Figure 1—source data 3*). The median is plotted in all figures, with statistical data reported as the median, mean ± SEM.

The online version of this article includes the following figure supplement(s) for figure 1:

**Source data 1.** Complete data for panel G.

*Figure 1 continued on next page*

*Figure 1 continued*

**Source data 2.** Western for panel G.

**Source data 3.** Western region used for panel G.

has been suggested to inhibit SV SNARE complex formation by competing for t-SNARE binding with the v-SNARE Syb2. To determine if *Drosophila* Tomosyn plays a similar role in negatively regulating SNARE complex formation, SDS-resistant SNARE complex (7S complex) abundance was assayed. Western blots of control and *tomosyn^NA1* brain lysates with Syntaxin1A antisera demonstrated a 2.7-fold increase in SNARE complex levels in the absence of Tomosyn (*Figure 1G*), consistent with Tomosyn inhibition of SNARE assembly.

To characterize synaptic transmission at Tomosyn-deficient synapses, two-electrode voltage clamp (TEVC) recordings were performed at 3rd instar muscle 6 NMJs in larval segment A4. *Tomosyn^NA1* null hemizygous males displayed a 62 % increase in the amplitude of evoked excitatory junctional currents (eEJCs) in 0.3 mM extracellular $Ca^{2+}$, indicating Tomosyn suppresses neurotransmitter release (*Figure 2A–C*). A similar 51 % increase in evoked release was found in the *tomosyn^FS1* frameshift mutant (*Figure 2—figure supplement 1A-C*). Enhanced release in *tomosyn* mutants could result from a larger postsynaptic response to single SVs (quantal size) or fusion of a larger number of SVs per stimulus (quantal content). Quantal size as measured by miniature excitatory junctional current (mEJC) amplitude was unchanged in *tomosyn^NA1* (*Figure 2D–F*). Instead, *tomosyn* mutant terminals released 70 % more SVs across the active zone (AZ) population in response to a single stimulus (*Figure 2G and H*), with an average increase in quantal content from 84 SVs released in controls to 143 in *tomosyn^NA1*. In higher extracellular $Ca^{2+}$ (2.0 mM) saline, evoked responses in *tomosyn^NA1* remained larger and displayed a slower evoked charge transfer that resulted in a 43 % increase in EJC area (*Figure 2I–M*). The enhancement in delayed SV release is consistent with *Drosophila* RNAi knockdowns and *tomosyn* mutants in other species (*Chen et al., 2011*; *Gracheva et al., 2006*; *McEwen et al., 2006*; *Saki-saka et al., 2008*). *Tomosyn^NA1* mutants also showed a 3.5-fold increase in the rate of stimulation-independent spontaneous miniature release events (*Figure 2N and O*), a phenotype not reported in *C. elegans* or mammalian *Tomosyn-1* mutants though present in mammalian *Tomosyn-2* mutants (*Geerts et al., 2015*). To confirm the elevated mini frequency was not due to a second-site mutation, *tomosyn^NA1* mutants were crossed with a deficiency line (*Df(1)ED7161*) spanning the *tomosyn* locus. This allelic combination showed similar increases in spontaneous release (*Figure 2N and O*), as did the *tomosyn^FS1* allele (*Figure 2—figure supplement 1D, E*). *Tomosyn^NA1/Df(1)7,161* trans-heterozygous null females showed even larger evoked responses compared to *tomosyn^NA1* hemizygous males or controls (*Figure 2A–C*). Together with the reduction in homozygous female viability, sex-specific differences in Tomosyn function are likely to account for these enhanced phenotypes. *Tomosyn* null males were used for the remainder of the study to avoid phenotypic sex differences.

To test conservation of Tomosyn function, *Drosophila* Tomosyn or human *Tomosyn-1* transgenes were pan-neuronally expressed in the *tomosyn^NA1* background using the Gal4/UAS system. Both homologs rescued the increased evoked and spontaneous release phenotypes in *tomosyn^NA1* (*Figure 2P and Q*). Unexpectedly, rescue with human Tomosyn-1 suppressed SV release to below control levels and below rescue with *Drosophila* Tomosyn. Immunohistochemistry against a Myc epitope attached to the transgenic Tomosyn proteins revealed human Tomosyn-1 was more abundant in presynaptic terminals than *Drosophila* Tomosyn (*Figure 2R and S*), suggesting dosage-sensitive inhibition of SV fusion is likely to account for the enhanced suppression. Together, these data indicate Tomosyn suppresses both evoked and spontaneous SV release at *Drosophila* NMJs, with these properties retained in human Tomosyn-1.

## The C-terminal SNARE domain of Tomosyn is essential for release suppression while the N-terminal scaffold promotes SV enrichment

To identify critical domains within Tomosyn that mediate suppression of SV release, full-length or truncation mutants were expressed in *tomosyn^NA1* using the Gal4/UAS system (*Figure 3*). Both Tomosyn13A and 13B splice variants restored neurotransmitter release in *tomosyn^NA1* (*Figure 3A and B*). Eliminating the SNARE motif from either splice isoform abolished rescue, while expressing the Tomosyn SNARE domain alone only partially rescued enhanced release (*Figure 3A*). Although the SNARE motif

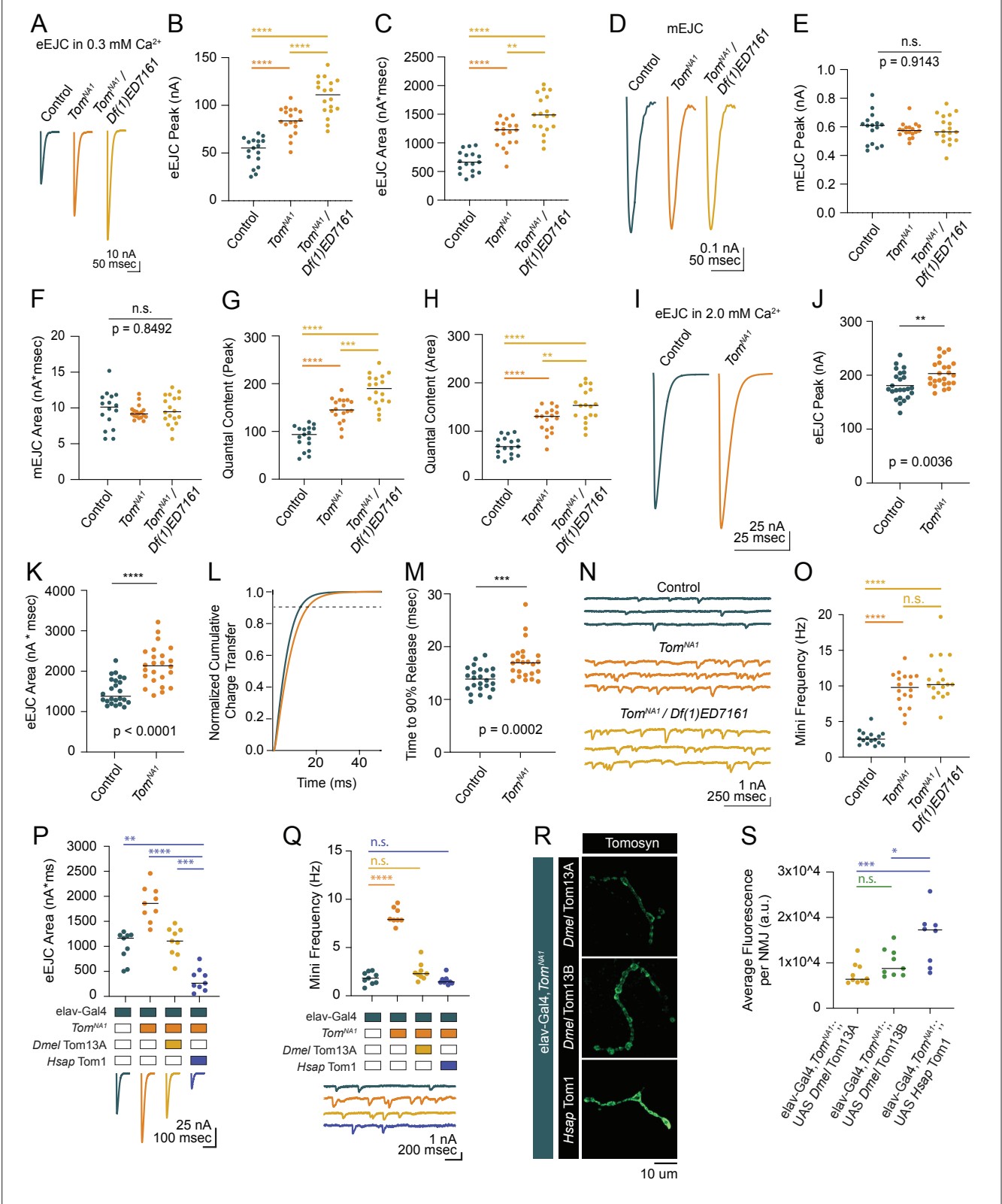

**Figure 2.** *Tomosyn* mutants show elevated evoked and spontaneous neurotransmitter release. (**A**) Average evoked eEJC traces in 0.3 mM Ca²⁺. (**B**) Quantification of average eEJC peak amplitude (nA) per neuromuscular junction (NMJ) in 0.3 mM Ca²⁺ (control: 55.3, 51.78±3.522, n=17; *tomosyn^NA1*: 83.74, 83.74±3.378, n=18; *tomosyn^NA1/Df(1)ED7161*: 111.0, 108.8±4.578, n=18; *p*<0.0001; ≥10 larvae per group). (**C**) Quantification of average eEJC area (nA*msec) per NMJ in 0.3 mM Ca²⁺ (control: 663.2, 670.7±45.60, n=17; *tomosyn^NA1*: 1228, 1167±56.66, n=18; *tomosyn^NA1/Df(1)ED7161*: 1488, 1499±78.35,

*Figure 2 continued on next page*

*Figure 2 continued*

n=18; *p*<0.0001; ≥10 larvae per group). (**D**) Average mEJC traces. (**E**) Quantification of average mEJC peak amplitude (nA) per NMJ (control: 0.6104, 0.5898±0.02706, n=16; *tomosyn*[NA1]: 0.5985, 0.5771±0.01221, n=18; *tomosyn*[NA1]/*Df(1)ED7161*: 0.5657, 0.5846±0.02287, n=18; *p*=0.9143; ≥10 larvae per group). (**F**) Quantification of average mEJC area (nA*msecs) per NMJ (control: 10.12, 9.743±0.6477, n=16; *tomosyn*[NA1]: 9.172, 9.396±0.2328, n=18; *tomosyn*[NA1]/ *Df(1)ED7161*: 9.476, 9.697±0.4741, n=18; *p*=0.8496; ≥10 larvae per group). (**G**) Quantification of evoked quantal content in 0.3 mM $Ca^{2+}$ per NMJ calculated using peak EJC (control: 93.75, 87.79±5.971, n=17; *tomosyn*[NA1]: 145.1, 145.1±5.854 n=18; *tomosyn*[NA1]/*Df(1)ED7161*: 189.9, 186.2±7.831, n=18; *p*<0.0001; ≥10 larvae per group). (**H**) Quantification of evoked quantal content in 0.3 mM $Ca^{2+}$ per NMJ calculated using EJC area (control: 68.07, 68.84±4.680, n=17; *tomosyn*[NA1]: 130.7, 124.2±6.030, n=18; *tomosyn*[NA1]/*Df(1)ED7161*: 153.5, 154.6± 8.080, n=18; *p*<0.0001; ≥10 larvae per group). (**I**) Average eEJC traces in 2.0 mM $Ca^{2+}$. (**J**) Quantification of average eEJC peak amplitude (nA) per NMJ in 2.0 mM $Ca^{2+}$ (control: 174.4, 181.0±5.313, n=24; *tomosyn*[NA1]: 197.1, 203.2±4.948 n=24; p=0.0036; ≥18 larvae per group). (**K**) Quantification of average eEJC area (nA*msec) per NMJ in 2.0 mM $Ca^{2+}$ (control: 1372, 1496±66.60, n=24; *tomosyn*[NA1]: 2134, 2140±97.90, n=24; p<0.0001; ≥18 larvae per group). (**L**) Normalized cumulative charge transfer of average eEJC in 2.0 mM $Ca^{2+}$; dashed line represents 90% cumulative release. (**M**) Quantification of time (msec) when average eEJC reaches 90% charge transfer per NMJ in 2.0 mM $Ca^{2+}$ (control: 13.85, 13.79±0.4711, n=24; *tomosyn*[NA1]: 16.95, 17.19±0.7025, n=24; p=0.0002; ≥18 larvae per group). (**N**) Representative mEJC traces. (**O**) Quantification of mEJC frequency (Hz) per NMJ (control: 2.547, 2.701±0.2436, n=16; *tomosyn*[NA1]: 9.783, 9.522±0.5590, n=18; *tomosyn*[NA1]/*Df(1)ED7161*: 10.19, 10.97±0.7395, n=18; *p*< 0.0001; ≥10 larvae per group). (**P**) Quantification of evoked peak current amplitude (nA) in 0.3 mM $Ca^{2+}$ in controls, *tomosyn*[NA1] mutants and *tomosyn*[NA1] mutants rescued with *Drosophila* (*Dmel* Tom13A) or human (*Hsap* Tom1) *tomosyn* (elav-Gal4: 1165, 996.6±101.7, n=9; elav-Gal4,*tomosyn*[NA1]: 1860, 1856±117.2, n=9; elav-Gal4,*tomosyn*[NA1]>UAS-*Drosophila* Tom13A: 1106, 1093±96.98, n=9 NMJs; elav-Gal4,*tomosyn*[NA1]>UAS-Human Tom1: 262.7, 330 ± 73.47, n = 9; *P* < 0.0001; ≥ 5 larvae per group). (**Q**) Quantification of mEJC rate (Hz) (elav-Gal4: 1.833, 1.836 ± 0.2098, n = 9; elav-Gal4,*tomosyn*[NA1]: 7.901, 8.268 ± 0.3066, n = 9; elav-Gal4,*tomosyn*[NA1]> UAS-*Dmel*Tom13A: 2.300, 2.497 ± 0.3029, n = 9; elav-Gal4,*tomosyn*[NA1]> UAS-*Hsap*Tom1: 1.438, 1.605 ± 0.1487, n = 9; p < 0.0001; ≥ 5 larvae per group). (**R**) Representative confocal images of Myc-tagged *Drosophila* (*Dmel* Tom13A and *Dmel* Tom13B) and human (*Hsap* Tom1) Tomosyn rescue constructs at 3[rd] instar NMJs. (**S**) Quantification of fluorescence intensity (arbitrary fluorescence units) of Myc-tagged Tomosyn rescue constructs (elav-Gal4,*tomosyn*[NA1]> UAS *Dmel* Tom13A: 6391, 7437 ± 742.9, n = 10; elav-Gal4,*tomosyn*[NA1]> UAS-*Dmel*Tom13B: 8764, 10,003 ± 1013,, n = 9; elav-Gal4,*tomosyn*[NA1]> UAS *Hsap* Tom1: 17,253, 15,528 ± 2141,, n = 8; p = 0.001; ≥ 6 larvae per group). Complete data for this figure provided in ***Figure 2—source data 1***.

The online version of this article includes the following figure supplement(s) for figure 2:

**Source data 1.** Source data for ***Figure 2***.

**Figure supplement 1.** *Tomosyn*[FS1] null mutants display elevated evoked and spontaneous neurotransmitter release.

**Figure supplement 1—source data 1.** Source data for ***Figure 2—figure supplement 1***.

is necessary for Tomosyn function, the failure of the SNARE-only construct to fully rescue release defects suggests the scaffold domain also contributes. The scaffold domain could be independently required or act together with the SNARE motif to inhibit fusion. Co-expression of the scaffold and SNARE domains as independent transgenes failed to reconstitute full-length Tomosyn function, indicating these domains must be linked and act cooperatively to decrease SV release (***Figure 3A***).

To determine whether the N-terminal scaffold domain regulates Tomosyn localization, anti-Myc immunocytochemistry was performed on the panel of Myc-tagged rescue constructs. Full-length Tomosyn was present throughout the periphery of presynaptic boutons as observed for other SV proteins (***Figure 3C and D***), consistent with the presence of Tomosyn on SVs in *C. elegans* and mammals (***Geerts et al., 2017***; ***McEwen et al., 2006***). Both Tomosyn13A and 13B co-localized with the SV protein Synapsin to a greater extent than with the neuronal plasma membrane marker anti-HRP (***Figure 3C and D***). Tomosyn13A and the 13 A scaffold domain alone (Tom13A-ΔSNARE) showed similar SV enrichment, suggesting the SNARE motif is dispensable for SV localization. The Tomosyn13B scaffold domain (Tom13B-ΔSNARE) was slightly less efficient at localizing this isoform to SV rich sites. In contrast to the scaffold domain, the Tomosyn SNARE motif alone showed reduced co-localization with SVs. Together, these data indicate the scaffold domain functions to enhance Tomosyn SV localization.

To determine whether Tomosyn bidirectionally modulates SV release, the protein was overexpressed in a wildtype background. Full-length Tomosyn13A suppressed evoked and spontaneous release by 33% and 40%, respectively (***Figure 3—figure supplement 1A,B***). Tomosyn13B overexpression was less effective at reducing release, although the 13B scaffold alone modestly decreased mini frequency (***Figure 3—figure supplement 1B***). Overexpression of the mammalian Tomosyn scaffold alone reduces SV fusion (***Yamamoto et al., 2010***; ***Yizhar et al., 2007***; ***Yizhar et al., 2004***), suggesting the Tomosyn13B scaffold may have similar properties. Overexpression of the remaining Tomosyn truncation mutants, including the SNARE motif alone, failed to inhibit evoked or spontaneous release. These data indicate Tomosyn13A acts as a bidirectional modulator of presynaptic output and requires both the scaffold and SNARE domains to fully regulate SV release.

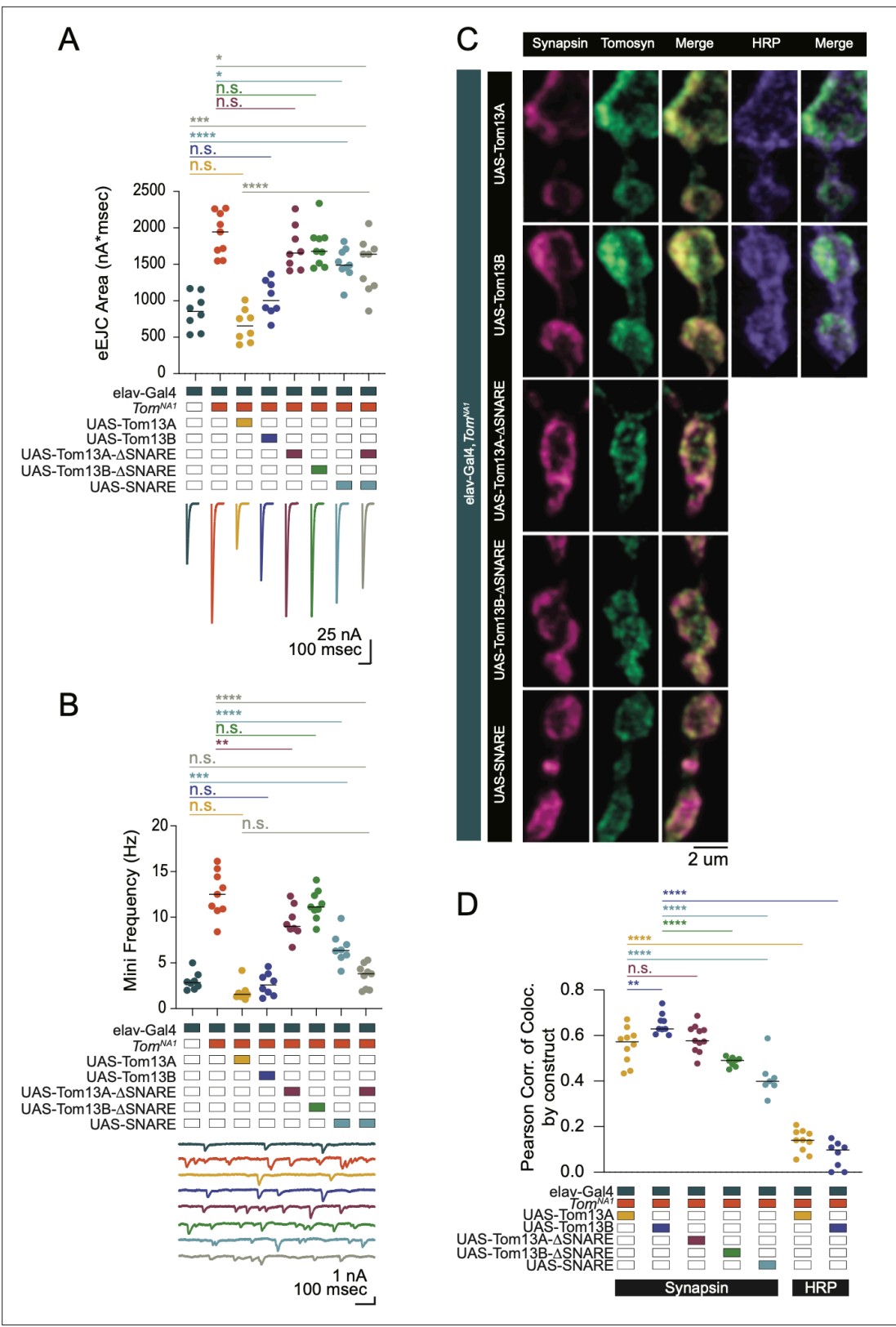

**Figure 3.** The Tomosyn SNARE domain mediates release suppression and the scaffold promotes SV enrichment. (**A**) Quantification of evoked eJC area (nA*msec) in Tomosyn rescue lines in 0.3 mM Ca²⁺ (elav-Gal4: 853.9, 852.3 ± 86.62, n = 8; elav-Gal4,*tomosyn*$^{NA1}$: 1942, 1915 ± 98.61, n = 9; elav-Gal4,*tomosyn*$^{NA1}$> UAS-Tom13A: 655.2, 662.2 ± 79.03, n = 8; elav-Gal4,*tomosyn*$^{NA1}$> UAS-Tom13B: 1004, 1037 ± 86.03, n = 8; elav-Gal4,*tomosyn*$^{NA1}$> UAS-Tom13A-ΔSNARE: 1656, 1726 ± 107.5, n = 8; elav-Gal4,*tomosyn*$^{NA1}$> UAS-Tom13B-ΔSNARE: 1678, 1742 ± 92.94, n = 9; elav-Gal4,*tomosyn*$^{NA1}$>

*Figure 3 continued on next page*

*Figure 3 continued*

UAS-SNARE: 1488, 1508 ± 71.46, n = 9; elav-Gal4,*tomosyn*[NA1]> UAS-SNARE,UAS-Tom13A: 1639, 1484 ± 124.5, n = 9; *P* < 0.0001; Šidak's multiple comparisons test, p-values indicated in figure; ≥ 4 larvae per group). (**B**) Quantification of mEJC rate (Hz) (elav-Gal4: 2.850, 3.000 ± 0.3429, n = 8; elav-Gal4,*tomosyn*[NA1]: 12.50, 12.54 ± 0.8283, n = 9; elav-Gal4,*tomosyn*[NA1]> UAS-Tom13A: 1.550, 1.817 ± 0.3511, n = 8; elav-Gal4,*tomosyn*[NA1]> UAS-Tom13B: 2.600, 2.671 ± 0.4371, n = 8; elav-Gal4,*tomosyn*[NA1]> UAS-Tom13A-ΔSNARE: 8.983, 9.467 ± 0.6319, n = 8; elav-Gal4,*tomosyn*[NA1]> UAS-Tom13B-ΔSNARE: 11.13, 11.34 ± 0.5356, n = 9; elav-Gal4,*tomosyn*[NA1]> UAS-SNARE: 6.367, 6.596 ± 0.5937, n = 8; elav-Gal4,*tomosyn*[NA1]> UAS-SNARE,UAS-Tom13AΔSNARE: 3.815, 3.571 ± 0.4309, n = 9; p < 0.0001; Šidak's multiple comparisons test, *p*-values indicated in figure; ≥ 4 larvae per group). (**C**) Representative confocal images of NMJs immunostained for Tomosyn (anti-Myc), Synapsin (3C11) and HRP in *tomosyn* rescue lines (full-length Tomosyn 13 A and 13B: UAS-Tom13A and UAS-Tom13B; SNARE deletions of Tomosyn 13 A and 13B: UAS-Tom13A-ΔSNARE and UAS-Tom13B-ΔSNARE; SNARE domain alone: UAS-SNARE). (**D**) Pearson correlation of co-localization between Tomosyn rescue constructs and Synapsin (elav-Gal4,*tomosyn*[NA1]> UAS-Tom13A: 0.05725, 0.5559 ± 0.02471, n = 10; elav-Gal4,*tomosyn*[NA1]> UAS-Tom13B: 0.6290, 0.6509 ± 0.01516, n = 9; elav-Gal4,*tomosyn*[NA1]> UAS-Tom13A-ΔSNARE: 0.5770, 0.5869 ± 0.01794, n = 11; elav-Gal4,*tomosyn*[NA1]> UAS-Tom13B-ΔSNARE: 0.4905, 0.4850 ± 0.007283, n = 8; elav-Gal4,*tomosyn*[NA1]> UAS-SNARE: 0.3990, 0.4161 ± 0.03189, n = 7) and between Tomosyn rescue constructs and HRP (elav-Gal4,*tomosyn*[NA1]> UAS-Tom13A: 0.1405, 0.1372 ± 0.01571, n = 10; elav-Gal4,*tomosyn*[NA1]> UAS-Tom13B: 0.09780, 0.07658 ± 0.02059, n = 8; p < 0.0001, Šidak's multiple comparisons test, p-values indicated in figure; ≥ 6 larvae per group). Complete data for this figure provided in *Figure 3—source data 1*.

The online version of this article includes the following figure supplement(s) for figure 3:

**Source data 1.** Source data for *Figure 3*.

**Figure supplement 1.** Tomosyn13A bidirectionally modulates evoked and spontaneous SV release.

**Figure supplement 1—source data 1.** Source data for *Figure 3—figure supplement 1*.

## Tomosyn restricts SV release in a Ca²⁺- and Synaptotagmin-independent manner

Beyond its role as a decoy SNARE, Tomosyn has been suggested to decrease release by binding to the Ca²⁺ sensor Synaptotagmin 1 (Syt1) and reducing its ability to activate fusion (*Yamamoto et al., 2010*). If a Syt1/Tomosyn interaction mediates release inhibition in *Drosophila*, loss of Syt1 should eliminate Tomosyn's ability to decrease SV fusion. To test this model, neurotransmitter release in *tomosyn; syt1* double null mutants (*tomosyn*[NA1]; *syt1*[AD4/N13]) was characterized. Most of the evoked response in *tomosyn*[NA1] was Syt1-dependent, as double mutants had a large reduction in evoked release compared to controls (*Figure 4A*). However, *tomosyn;syt1* evoked responses were 86 % larger than those of *syt1* mutants alone (*Figure 4A and B*), indicating Tomosyn inhibits release independent of Syt1. *Syt1* mutants also show increases in the slower asynchronous phase of evoked fusion (*Jorquera et al., 2012*; *Yoshihara and Littleton, 2002*). Asynchronous release was increased 1.7-fold in *tomosyn;syt1* double mutants compared to *syt1* alone (*Figure 4C–F*), indicating Tomosyn reduces both synchronous and asynchronous SV fusion. Similar to Tomosyn suppression of spontaneous release at wild-type synapses, the elevated mini frequency normally observed in *syt1* was enhanced 2.8-fold in *tomosyn;syt1* double mutants (*Figure 4G and H*). Together, these data indicate Syt1 and Tomosyn regulate evoked and spontaneous SV fusion through independent mechanisms.

Another member of the Synaptotagmin family, Syt7, regulates evoked release by controlling the size and usage of the fusogenic SV pool in *Drosophila* (*Guan et al., 2020*). Like *tomosyn*, Syt7 null mutants (*syt7*[M1]) show increased quantal content, suggesting Syt7 and Tomosyn may restrict SV availability and fusion via a shared pathway. To test this hypothesis, we generated and characterized *tomosyn*[NA1];;;*syt7*[M1] double mutants. Both evoked release and mini frequency at *syt7*[M1] mutant NMJs was enhanced by loss of Tomosyn (*Figure 4I–L*), indicating the proteins act through independent mechanisms to negatively regulate SV fusion. In addition, increased evoked release in *tomosyn;;;syt7* double mutants indicate presynaptic output can still be enhanced beyond that observed in the absence of Tomosyn alone.

We next assayed if Tomosyn inhibition of SV release is Ca²⁺-sensitive by recording evoked responses across a range of extracellular [Ca²⁺]. Loss of Tomosyn enhanced release across the entire Ca²⁺ range but did not alter the Ca²⁺ cooperativity of release (*Figure 4—figure supplement 1A, B*). Paired-pulse stimulation in Ca²⁺ conditions that matched first pulse EJC amplitudes between *tomosyn*[NA1] and controls revealed facilitation was also preserved in the absence of Tomosyn (*Figure 4—figure supplement 1C, D*). At interstimulus intervals (ISI) of 25 and 50 msec, *tomosyn* mutants displayed enhanced paired-pulse facilitation (PPF). At 75 ms ISI, PPF is preserved but not significantly enhanced (*Figure 4—figure supplement 1E, F*). The preservation of PPF suggests Tomosyn is unlikely to reduce fusogenicity of individual SVs. Given *tomosyn* mutants do not decrease the Ca²⁺ dependence of fusion

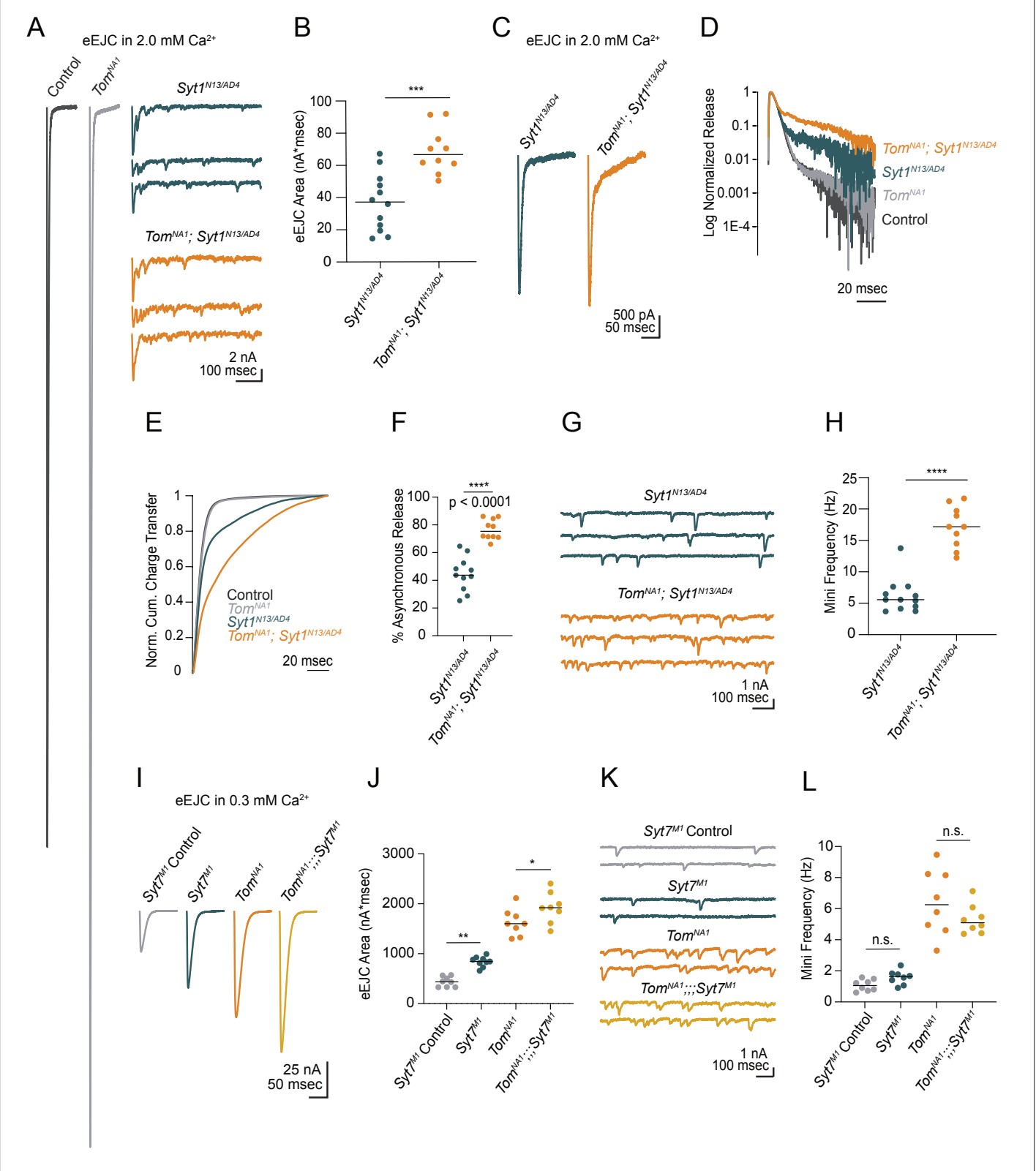

**Figure 4.** Tomosyn inhibits release independent of Synaptotagmin 1 and 7. (**A**) Average evoked response in 2.0 mM Ca²⁺ for control and *tomosyn*^NA1 (left) compared to representative traces of *syt1* nulls (*Syt1*^N13/AD4) and *tomosyn/syt1* double mutants (*Tom*^NA1;*Syt1*^N13/AD4, right). (**B**) Quantification of average evoked response area (nA*msec) per NMJ in 2.0 mM Ca²⁺ (*syt1*^N13/AD4: 37.22, 37.13 ± 5.139, n = 12; *tomosyn*^NA1/*syt1*^N13/AD4: 66.59, 69.05 ± 4.471, n = 10; p = 0.0002; ≥ 5 larvae per group). (**C**) Average EJC response in 2.0 mM Ca²⁺. (**D**) Log normalized average evoked response in 2.0 mM Ca²⁺. (**E**)

*Figure 4 continued on next page*

Figure 4 continued

Normalized cumulative charge transfer for the average evoked response in 2.0 mM Ca$^{2+}$. (F) The slow component of evoked release was identified by fitting a double logarithmic function to the average normalized cumulative charge transfer per NMJ in 2.0 mM Ca$^{2+}$ and plotted as a percent of total charge transfer (syt1$^{N13/AD4}$: 43.69, 44.55 ± 3.717, n = 11; tomosyn$^{NA1}$/syt1$^{N13/AD4}$: 75.38, 76.72 ± 2.295, n = 10; p < 0.0001; ≥ 5 larvae per group). (G) Representative mEJC traces. (H) Quantification of mEJC rate (Hz) per NMJ (syt1$^{N13/AD4}$: 5.567, 6.192 ± 0.7904, n = 12; tomosyn$^{NA1}$, syt1$^{N13/AD4}$: 17.17, 17.17 ± 1.034, n = 10; p < 0.0001; ≥ 5 larvae per group). (I) Average evoked response in 0.3 mM Ca$^{2+}$ of control (Syt7$^{M1}$ control), syt7 null (Syt7$^{M1}$), tomosyn null (tomosyn$^{NA1}$), and tomosyn/syt7 double null (tomosyn$^{NA1}$;;;Syt7$^{M1}$). (J) Quantification of average evoked response area (nA*msec) per NMJ in 0.3 mM Ca$^{2+}$ (control: 437.4, 437.4 ± 36.11, n = 8; syt7$^{M1}$: 844.6, 840.4 ± 33.77, n = 9; tomosyn$^{NA1}$: 1602, 1627 ± 94.19, n = 8; tomosyn$^{NA1}$, syt7$^{M1}$: 1920, 1923 ± 108.4, n = 8; p < 0.0001; ≥ 5 larvae per group). (K) Representative mEJC traces. (L) Quantification of mEJC rate per NMJ (Hz) (control: 1.056, 1.070 ± 0.1290, n = 8; syt7$^{M1}$: 1.617, 1.569 ± 0.161, n = 8; tomosyn$^{NA1}$: 6.256, 6.404 ± 0.7475, n = 8; tomosyn$^{NA1}$, syt7$^{M1}$: 5.092, 5.304 ± 0.3292, n = 8; p < 0.0001; ≥ 5 larvae per group). Complete data for this figure provided in Figure 4—source data 1.

The online version of this article includes the following figure supplement(s) for figure 4:

Source data 1. Source data for Figure 4.

Figure supplement 1. Tomosyn inhibits release in a Ca$^{2+}$-independent mechanism.

Figure supplement 1—source data 1. Source data for Figure 4—figure supplement 1.

or PPF, Tomosyn inhibits SV release through a Ca$^{2+}$-independent mechanism. Loss of Tomosyn also leads to enhanced synchronous and asynchronous release, together with elevated rates of spontaneous SV fusion. These data indicate Tomosyn controls SV supply independent of the specific route for SV release, likely by sequestering free t-SNAREs to reduce fusogenic SNARE complex formation.

## Tomosyn mutants have more docked SVs at individual release sites

The enhanced evoked response in tomosyn mutants could reflect an increased number of AZs per NMJ, a higher number of docked SVs per AZ, or an increase in individual SV fusogenicity. Given tomosyn mutants display enhanced PPF, increased SV fusogenicity is unlikely. To determine if AZ number or SV docking is increased, immunocytochemistry and transmission electron microscopy (TEM) were used to characterize the morphology and ultrastructure of tomosyn$^{NA1}$ NMJs. Immunostaining for the AZ scaffold protein Bruchpilot (BRP) demonstrated AZ number was unchanged at tomosyn synapses (Figure 5A and B). Additionally, BRP abundance at individual AZs was not altered (Figure 5C), indicating release site number and AZ scaffold accumulation are not affected. To further probe NMJ morphology, bouton size and number were analyzed in tomosyn mutants. Despite a slightly smaller bouton area at Ib NMJs, total NMJ area was unchanged due to a mild increase in the number of boutons per NMJ (Figure 5D–F). Is terminals showed no morphological differences from controls (Figure 5D and E). Immunostaining for Syt1 revealed total SV abundance was not altered in tomosyn mutants (Figure 5G–H). Together, these data indicate morphological defects or AZ number are unlikely to account for enhanced SV release.

TEM was used to determine whether enhanced SV docking contributes to increased SV release in tomosyn mutants. Despite no gross changes to bouton ultrastructure (Figure 6A), Ib terminals had a 52 % increase in the number of docked SVs per AZ in tomosyn$^{NA1}$ (Figure 6B and C). Increased docking was observed over the entire length of the AZ, with no change in the absolute fraction of docked SVs along the 400 nm trajectory from the electron dense T-bar center (Figure 6D and E). The average distance between neighboring SVs was also unchanged (Figure 6F), suggesting SV clustering is not altered. Tomosyn mutants also showed a larger number of SVs within 100 and 150 nm concentric circles positioned over the AZ center (Figure 6G and H). Average SV diameter (Figure 6I) and SV density were unchanged (Figure 6J), indicating Tomosyn does not affect SV formation or total SV number. Together, these data suggest Tomosyn suppresses release by decreasing SV availability at AZs, with enhanced SV docking occurring in the absence of the protein.

## Tomosyn decreases the rate of SV usage during high-frequency stimulation

Endogenous activity at larval NMJs is controlled by central pattern generators (CPGs) that trigger intermittent high frequency motoneuron bursting (5–40 Hz) to drive locomotion (Jan and Jan, 1976; Lu et al., 2016; Pulver et al., 2015). To examine how elevated release in tomosyn mutants change during different rates of neuronal firing, synaptic output was compared between low (0.33 Hz) and high (10 Hz) frequency stimulation in 2 mM extracellular Ca$^{2+}$. Consistent with the enhanced single

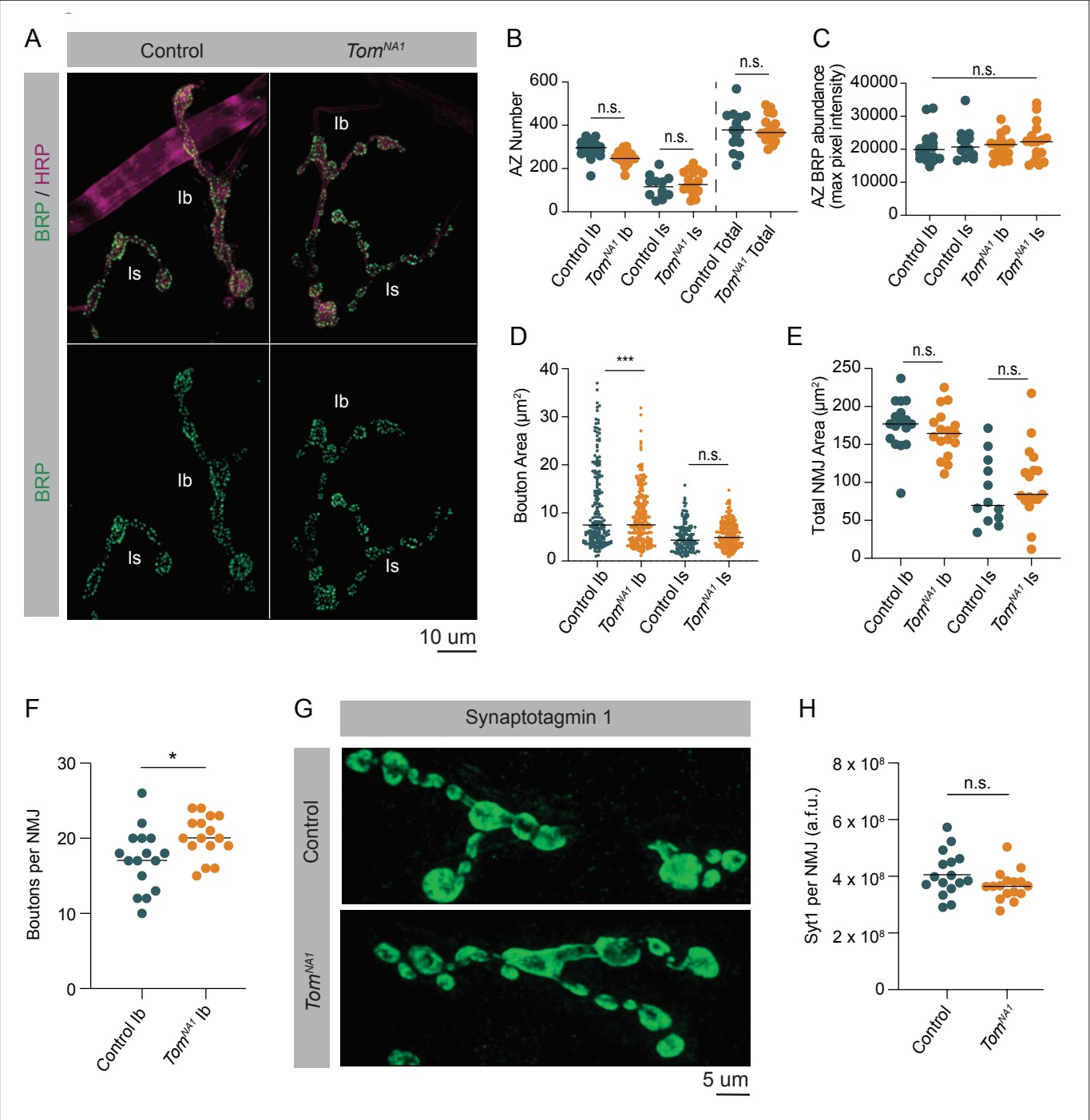

**Figure 5.** Loss of Tomosyn does not affect AZ number, NMJ area or SV abundance. (**A**) Representative confocal images of immunohistochemistry against BRP (nc82) and neuronal membranes (anti-HRP). (**B**) Quantification of AZ number per muscle 4 NMJ for Ib, Is and both (control, Ib: 296.0, 288.2 ± 10.71, n = 17; control, Is: 115.5, 115.8 ± 14.97, n = 12; *tomosyn*[NA1], Ib: 246.0, 249.7 ± 7.824, n = 17; *tomosyn*[NA1], Is: 126.0, 134.1 ± 12.52, n = 17; control, total: 378.0, 374.8 ± 25.07, n = 14; *tomosyn*[NA1], total: 366.0, 383.9 ± 14.74, n = 17; p = 0.0001; ≥ 7 larvae per group). (**C**) Quantification of average BRP abundance per AZ per muscle 4 NMJ, measured as average of maximum pixel intensity of each BRP puncta in arbitrary fluorescence intensity units (control, Ib: 19911, 20722 ± 1210,, n = 17; control, Is: 20682, 21733 ± 1448,, n = 12; *tomosyn*[NA1], Ib: 21430, 20681 ± 895.7, n = 17; *tomosyn*[NA1], Is: 22275, 22172 ± 1372,, n = 17; p = 0.7654; ≥ 7 larvae per group). (**D**) Quantification of average bouton size (um²) per muscle 4 NMJ measured as the HRP positive area for each bouton along the arbor (control, Ib: 7.489, 12.16 ± 1.013, n = 195; control, Is: 4.342, 4.974 ± 0.2488, n = 140; *tomosyn*[NA1], Ib: 7.508, 8.953 ± 0.3671, n = 241; *tomosyn*[NA1], Is: 4.873, 5.413 ± 0.1777, n = 229; p = 0.7654; ≥ 7 larvae per group). (**E**) Quantification of muscle 4 NMJ area (um²) measured as HRP positive area (control, Ib: 176.9, 176.0 ± 8.056, n = 17; control, Is: 69.76, 88.9 ± 12.89, n = 12; *tomosyn*[NA1], Ib: 164.5, 164.7 ± 7.527, n = 17; *tomosyn*[NA1], Is: 84.32, 99.69 ± 11.80, n = 17; p < 0.0001; ≥ 7 larvae per group). (**F**) Quantification of muscle four bouton number per Ib motoneuron

*Figure 5 continued on next page*

Figure 5 continued

(control: 17.50, 17.19 ± 1.030, n = 16; tomosyn$^{NA1}$: 20, 20.19 ± 0.7025, n = 16; p = 0.0225 ≥ 7 larvae per group). (**G**) Representative NMJs stained with Syt1 antisera. (**H**) Quantification of Syt1 expression (sum of arbitrary fluorescence units) per Ib motoneuron (control: 3.913*10^8, 4.083*10^8 ± 0.949* 10^8, n = 16; tomosyn$^{NA1}$: 3.3900* 10^8, 3.6713*10^8 ± 0.1297*10^8, n = 16; p = 0.0892 ≥ 7 larvae per group). Complete data found in **Figure 5—source data 1**.

The online version of this article includes the following figure supplement(s) for figure 5:

**Source data 1.** Source data for **Figure 5**.

eEJC phenotype, low frequency stimulation (0.33 Hz) resulted in a 4.3-fold increase in the EJC area of the first evoked response, followed by subsequent depression that stabilized at a 1.4-fold increased quantal content per action potential in tomosyn$^{NA1}$ (**Figure 7A and B**). At 10 Hz stimulation, tomosyn mutants displayed more robust synaptic depression, with quantal content quickly dropping below control levels (**Figure 7A and B**). Control NMJs had lower initial quantal content at 10 Hz and showed a gradual depression in release that was eventually equivalent to synaptic output of tomosyn$^{NA1}$ terminals by the 30th stimulus (**Figure 7C**). The size of the immediately releasable SV pool (IRP), approximated by the cumulative number of quanta released within 30 stimuli, showed no difference between tomosyn$^{NA1}$ and controls (**Figure 7D**). However, the depression rate was dramatically enhanced in tomosyn (**Figure 7E**), indicating Tomosyn normally restricts release from the IRP. To approximate the size of the larger readily releasable SV pool (RRP) and the SV recycling rate, 10 Hz stimulation was continued for 1,500 stimuli to reach steady state where the number of SVs released equals the number of recycled SVs (**Thanawala and Regehr, 2016**). The recycling rate in tomosyn$^{NA1}$ mutants was not significantly different from controls, though the RRP size was increased by 42 % (**Figure 7—figure supplement 1A-D**). Together these data indicate Tomosyn is required to support sustained release by limiting the number of fusogenic SVs.

## Tomosyn differentially regulates SV release from tonic Ib and phasic Is motoneurons

Tonic Ib and phasic Is motoneurons differ in their ability to sustain release during stimulus trains, with Ib synapses showing continued release and Is terminals displaying high initial $P_r$ and rapid depression (**Aponte-Santiago and Littleton, 2020**; **Lu et al., 2016**). Given the phasic release character of tomosyn mutant synapses (**Figure 7A–E**), we examined if Tomosyn differentially regulates release from Ib and Is motoneuron populations. To probe endogenous expression of Tomosyn, the GFP variant mClover3 was inserted into the tomosyn 13 A genomic locus (tomosyn$^{13A-Clover}$) using CRISPR (**Figure 1E**). Immunostaining for Tomosyn$^{13A-Clover}$ revealed a 2.1-fold enrichment of endogenous Tomosyn in Ib terminals relative to Is NMJs (**Figure 7F and G**). To determine whether this difference in expression resulted in functional changes in neurotransmitter release between the two classes of motoneurons, optogenetics was used to isolate Ib and Is evoked responses using motoneuron-specific Gal4 drivers to express UAS-channelrhodopsin2 (ChR2) (**Aponte-Santiago et al., 2020**; **Dawydow et al., 2014**; **Pérez-Moreno and O'Kane, 2019**). Optogenetic stimulation of Ib synapses in tomosyn$^{NA1}$ mutants showed a 3.8-fold increase in evoked EJC area (**Figure 7H and I**). In contrast, optogenetic stimulation of tomosyn$^{NA1}$ Is terminals revealed no differences in evoked output, indicating enhanced release in tomosyn mutants is solely contributed from increased SV fusion at Ib terminals. These data indicate higher expression of Tomosyn in Ib motoneurons results in greater intrinsic release suppression.

Is and Ib motoneuron populations also show stereotyped difference in single AZ $P_r$, with Is having intrinsically higher $P_r$ than Ib. To determine whether Tomosyn differentially regulates $P_r$, optical quantal analysis was performed in the tomosyn$^{FS1}$ null mutant. This mutant lacks the DsRed reporter cassette found in tomosyn$^{NA1}$ and has less background fluorescence during live imaging. To detect single SV release events at individual AZs, membrane-tethered GCaMP7s was expressed in postsynaptic muscles along with a tagged GluR subunit (GluRIIA-RFP) to identify individual PSDs as previously described (**Akbergenova et al., 2018**). Nerve stimulation in control animals indicated Is motoneurons showed a 2.4-fold higher average AZ $P_r$ (0.17 ± 0.007) than Ib motoneurons (0.07 ± 0.004). In contrast, tomosyn$^{FS1}$ mutants displayed higher $P_r$ at Ib AZs than Is due to increased Ib $P_r$ and no effect on Is $P_r$ (**Figure 7J–L**). Together, these data indicate Tomosyn suppresses release from tonic synapses and contributes to the intrinsic release differences between these motoneuron subclasses. Loss of

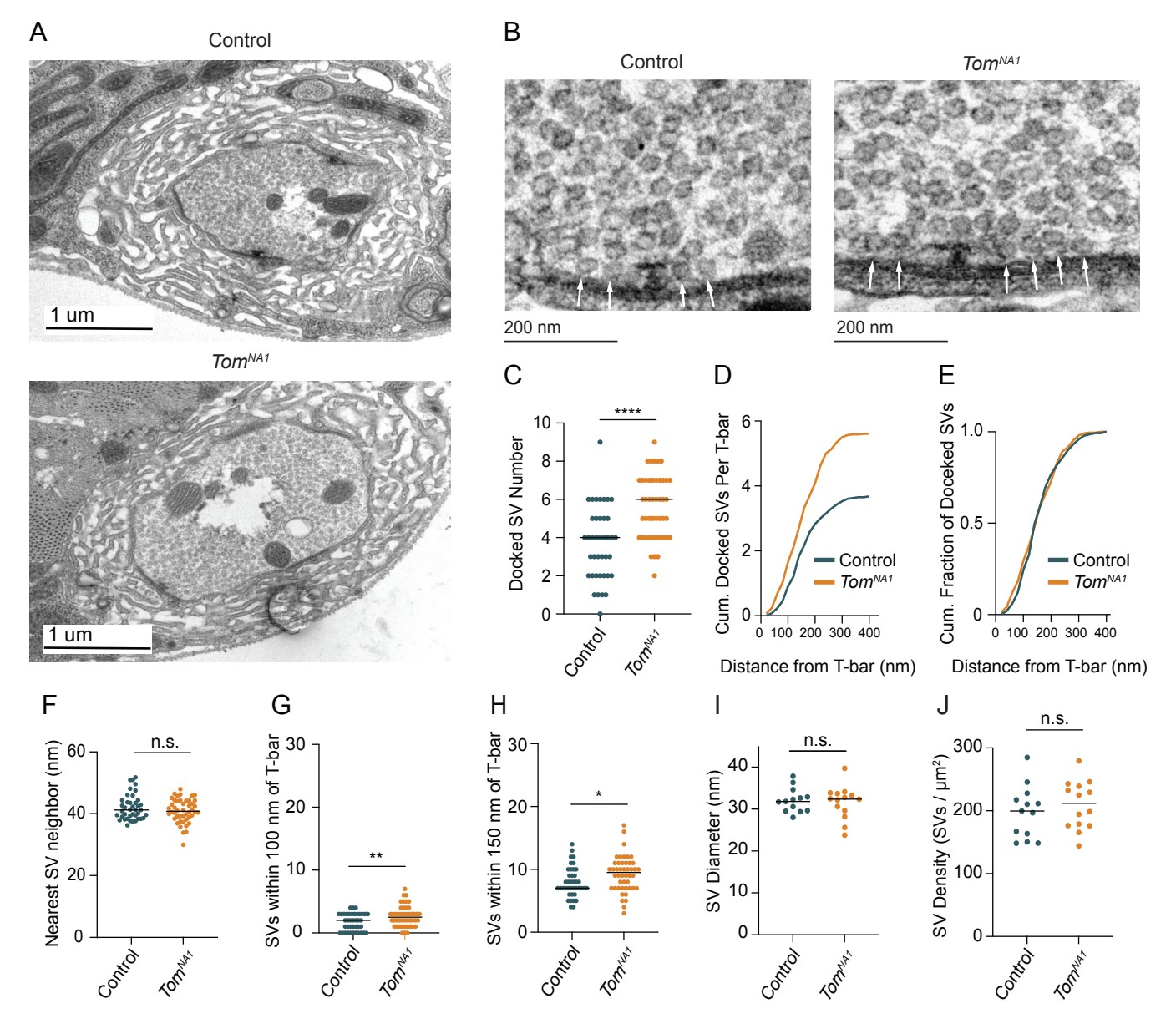

**Figure 6.** Tomosyn negatively regulates SV docking. (**A**) Representative TEM bouton cross-sections at Ib NMJs. (**B**) Representative micrographs of Ib AZs. Arrows indicate docked SVs. (**C**) Quantification of docked SV number along each AZ electron density (control: 4, 3.7 ± 0.3, n = 40 AZs; *tomosyn*[NA1]: 6, 5.609 ± 0.2437, n = 48 AZs; p < 0.0001; three larvae per group). (**D**) Average cumulative number of docked SVs at each distance from the T-bar center. (**E**) Docked SV distance from the AZ center, plotted as cumulative fraction of docked SVs at each distance from T-bar. (**F**) Quantification per micrograph of average distance (nm) from each SV to its nearest neighbor (control: 41.16, 42.02 ± 0.6476, n = 40 micrographs; *tomosyn*[NA1]: 40.78 nm, 40.91 ± 0.5561, n = 46 micrographs; p = 0.1931; three larvae per group). (**G**) Quantification of SV number closer than 100 nm to the T-bar (control: 1, 1.075 ± 0.1535, n = 40 AZs; *tomosyn*[NA1]: 2, 1.739 ± 0.1927, n = 46 AZs; p = 0.0099; three larvae per group). (**H**) Quantification of SV number closer than 150 nm to the T-bar (control: 7, 7.950 ± 0.3772, n = 40 AZs; *tomosyn*[NA1]: 9.5, 9.261 ± 0.4164, n = 46 AZs; p = 0.0236; three larvae per group). (**I**) Quantification of average SV diameter (nm) per micrograph (control: 31.81, 32.11 ± 0.7935, n = 13 boutons; *tomosyn*[NA1]: 32.40, 31.59 ± 1.050, n = 14 boutons; p = 0.7005; three larvae per group). (**J**) Quantification of average SV density per bouton area (SVs/um²) per micrograph (control: 199.6, 197.9 ± 11.52, n = 13 boutons; *tomosyn*[NA1]: 211.8, 209.2 ± 10.24, n = 14 boutons; p = 0.4690; three larvae per group). Complete data found in *Figure 6—source data 1*.

The online version of this article includes the following figure supplement(s) for figure 6:

**Source data 1.** Source data for *Figure 6*.

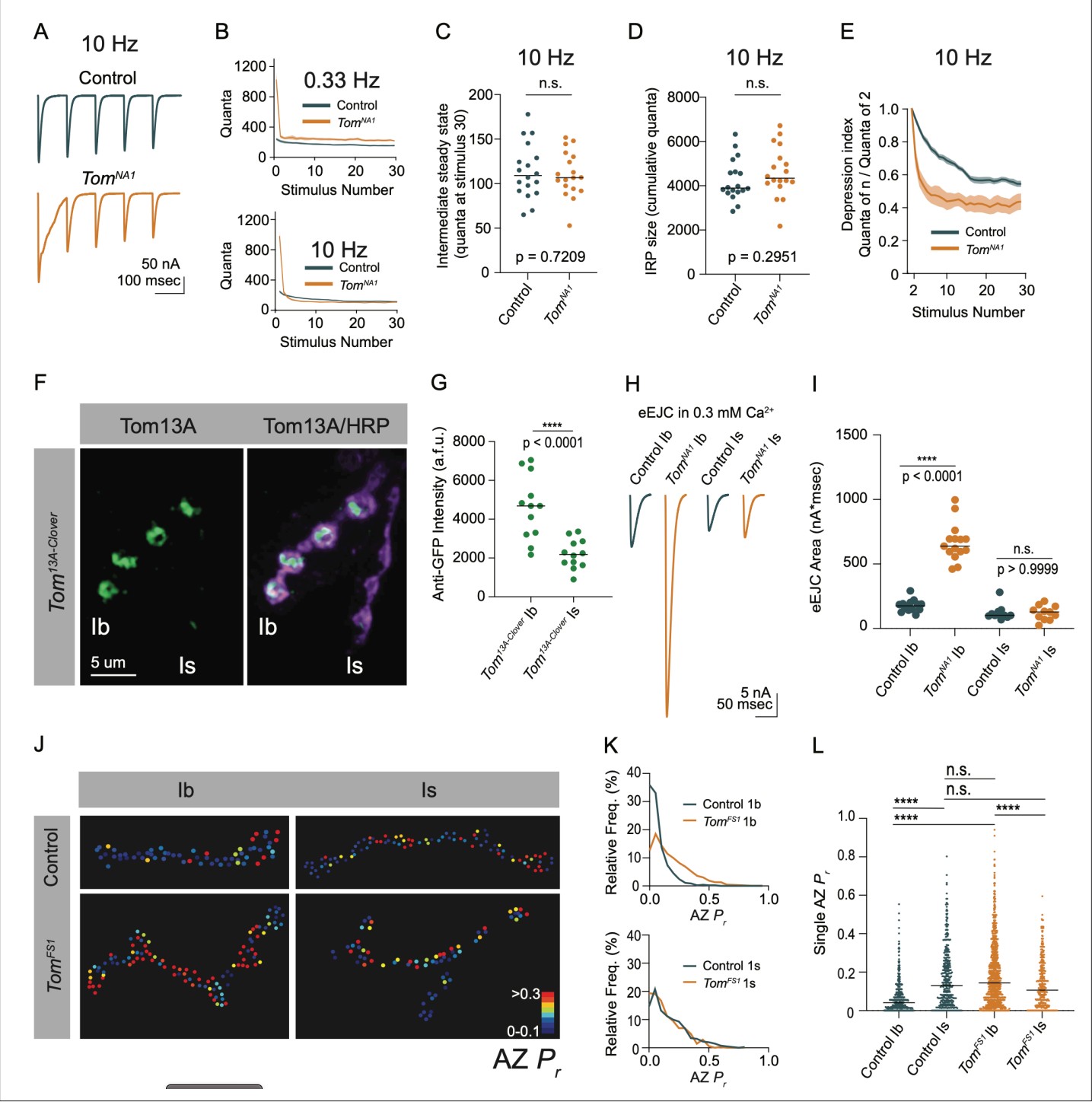

**Figure 7.** Tomosyn regulates tonic versus phasic release properties. (**A**) Average evoked response trains during 10 Hz stimulation in 2.0 mM Ca²⁺. Stimulus artifacts were removed for clarity. (**B**) Evoked quantal content in 2.0 mM Ca²⁺ (quanta) during a 0.33 Hz stimulus train (top) and during a 10 Hz stimulus train (bottom). Lines indicate mean values, with SEM noted by the shaded area (SEM is partly obscured in these plots by the line indicating the mean). (**C**) Quantification of evoked response size (quanta) at intermediate steady state, approximated as size of stimulus 30 following 10 Hz stimulation in 2.0 mM Ca²⁺ (control: 109.0, 113.8 ± 7.217, n = 18; *tomosyn^{NA1}*: 106.8, 8.110 ± 5.964, n = 18; p = 0.7209; ≥ 12 larvae per group). (**D**) Quantification of the immediately releasable pool size, approximated as the cumulative quanta released within 30 stimulations at 10 Hz in 2.0 mM Ca²⁺ (control: 3899, 4247 ± 219.5, n = 18; *tomosyn^{NA1}*: 4314, 4615 ± 268.1, n = 18; p = 0.2951; ≥ 12 larvae per group). (**E**) The depression index was calculated as the ratio of stimulus n to stimulus 2 during a 10 Hz train in 2.0 mM Ca²⁺. At stimulus 30, the depression index is: control: 0.5565, 0.5527 ± 0.1885, n = 18; *tomosyn^{NA1}*: 0.3961, 0.4277 ± 0.03628, n = 17 (p = 0.0039; ≥ 12 larvae per group). (**F**) Representative NMJ images of anti-GFP staining in *tomosyn^{13A-Clover}*.

*Figure 7 continued on next page*

Figure 7 continued

(**G**) Quantification of Tomosyn13A-Clover expression level (arbitrary fluorescence units) in Ib and Is terminals (*tomosyn^13A-Clover*, Ib: 4680, 4601 ± 475.1, n = 12; *tomosyn^13A-Clover*, Is: 2180, 2201 ± 215.7, n = 12; p < 0.0001; ≥ 4 larvae per group). (**H**) Average optically evoked responses from motoneurons expressing ChR2 with Gal4 drivers only in Ib (GMR94G06) or Is (GMR27F01). (**I**) Quantification of optically evoked response area (nA*msec) in Ib and Is (GMR94G06> UAS-ChR2: 175.2, 175.7 ± 12.31, n = 14; *tomosyn^NA1*, GMR94G06> UAS-ChR2: 638.0, 667.1 ± 37.91, n = 15; GMR27F01> UAS-ChR2: 101.9, 121.1 ± 17.05, n = 11; *tomosyn^NA1*, GMR27F01> UAS-ChR2: 128.7, 120.6 ± 17.01, n = 11; p < 0.0001; ≥ 5 larvae per group). (**J**) Representative maps of AZ $P_r$ at Ib or Is terminals in control or *tomosyn^FS1* mutants following optical quantal analysis. (**K**) Histogram of single AZ $P_r$ at Ib (top) and Is (bottom) NMJs. (**L**) Quantification of single AZ $P_r$ per motoneuron per genotype (the mean is plotted, control Ib: 0.04150, 0.06938 ± 0.003829, n = 463 AZs; control Is: 0.1295, 0.1664 ± 0.007488, n = 409 AZs; *tomosyn^FS1* Ib: 0.1434, 0.1846 ± 0.004917, n = 1,075 AZs; *tomosyn^FS1* Is: 0.1066, 0.1389 ± 0.006720, n = 346 AZs; p < 0.0001; ≥ 4 larvae per group). Complete data provided in *Figure 7—source data 1*.

The online version of this article includes the following figure supplement(s) for figure 7:

**Source data 1.** Source data for *Figure 7*.

**Figure supplement 1.** Steady-state recycling rate and RRP size in *tomosyn* mutants.

**Figure supplement 1—source data 1.** Source data for *Figure 7—figure supplement 1*.

Tomosyn in Ib neurons changes both the initial $P_r$ and short-term depression properties such that tonic Ib terminals display phasic release similar to Is motoneurons.

## Tomosyn is required for presynaptic homeostatic potentiation

Reductions in postsynaptic GluR function at *Drosophila* NMJs trigger a rapid and robust increase in presynaptic quantal content that homeostatically compensates for decreased quantal size (*Davis et al., 1998*; *Frank et al., 2006*; *Li et al., 2018*; *Petersen et al., 1997*). Given Tomosyn is a key regulator of quantal content, and prior data suggest PHP is more robust at tonic Ib synapses (*Newman et al., 2017*), we assayed if Tomosyn is required for PHP in tonic motoneurons. An allosteric inhibitor of *Drosophila* GluRs (Gyki) was used to acutely reduce quantal size and induce PHP as previously described (*Nair et al., 2020*). Following addition of Gyki into the extracellular saline, quantal size as measured by mini amplitude was reduced in both control and *tomosyn^NA1* mutants (*Figure 8A–C*). Mini frequency was not significantly changed following Gyki application, indicating spontaneous fusion events remained detectable (*Figure 8—figure supplement 1A, B*). Control animals compensated for the reduction in quantal size with a 62 % increase in quantal content that preserved the original evoked response amplitude (*Figure 8D–F*). In contrast, *tomosyn* NMJs showed no significant enhancement in quantal content after Gyki application, indicating PHP expression is impaired. Loss of PHP could result from an inability to support higher levels of release, or Tomosyn could be a key effector for PHP with post-translational modification decreasing its inhibitory function. To test if impaired PHP in *tomosyn* mutants is due to release saturation, the quantal content of potentiated NMJs in 0.35 mM extracellular [Ca$^{2+}$] was compared to non-potentiated NMJs at 1.5 mM [Ca$^{2+}$]. In elevated Ca$^{2+}$, quantal content was greater than after Gyki-induced potentiation in low Ca$^{2+}$ for controls (46 % increase) and *tomosyn^NA1* (61 % increase), indicating lack of potentiation in *tomosyn* is not due to release saturation (*Figure 8—figure supplement 1C*). Together with the observation that *tomosyn, syt7* double mutants displayed even higher levels of evoked release than *tomosyn* mutants alone in low Ca$^{2+}$ (*Figure 4J*), these data indicate Tomosyn is required for normal expression of Gyki-induced PHP and represents a key effector for enhancing presynaptic output during this form of plasticity.

Ib and Is motoneurons also differ in their ability to express PHP, with tonic Ib neurons showing more robust PHP in *GluRIIA* mutants (*Newman et al., 2017*). To monitor how differential Tomosyn expression in Ib and Is motoneurons affects expression of PHP in real time, optical quantal mapping was used to monitor AZ $P_r$ at individual release sites before and after acute Gyki application. Because Gyki reduces the fluorescent change (ΔF) from quantal release by decreasing postsynaptic Ca$^{2+}$ influx from GluRs (*Figure 8—figure supplement 1D*), transgenic animals expressing the more sensitive GCaMP variant GCaMP8s (*Zhang et al., 2020*) fused to a myristoylation domain for membrane tethering were generated to ensure SV release events could still be detected after Gyki application. Control Ib terminals showed a rapid and robust 1.8-fold increase in average AZ $P_r$ 15 min after Gyki incubation (*Figure 8G–J*). Enhanced SV release occurred across the majority of the AZ population. In addition, previously silent AZs were recruited during evoked stimulation following Gyki application (*Figure 8G*). In contrast to the robust effect at Ib synapses, Is terminals showed no significant change in AZ $P_r$ or recruitment of silent AZs following Gyki application (*Figure 8G and K–M*), indicating this form of PHP

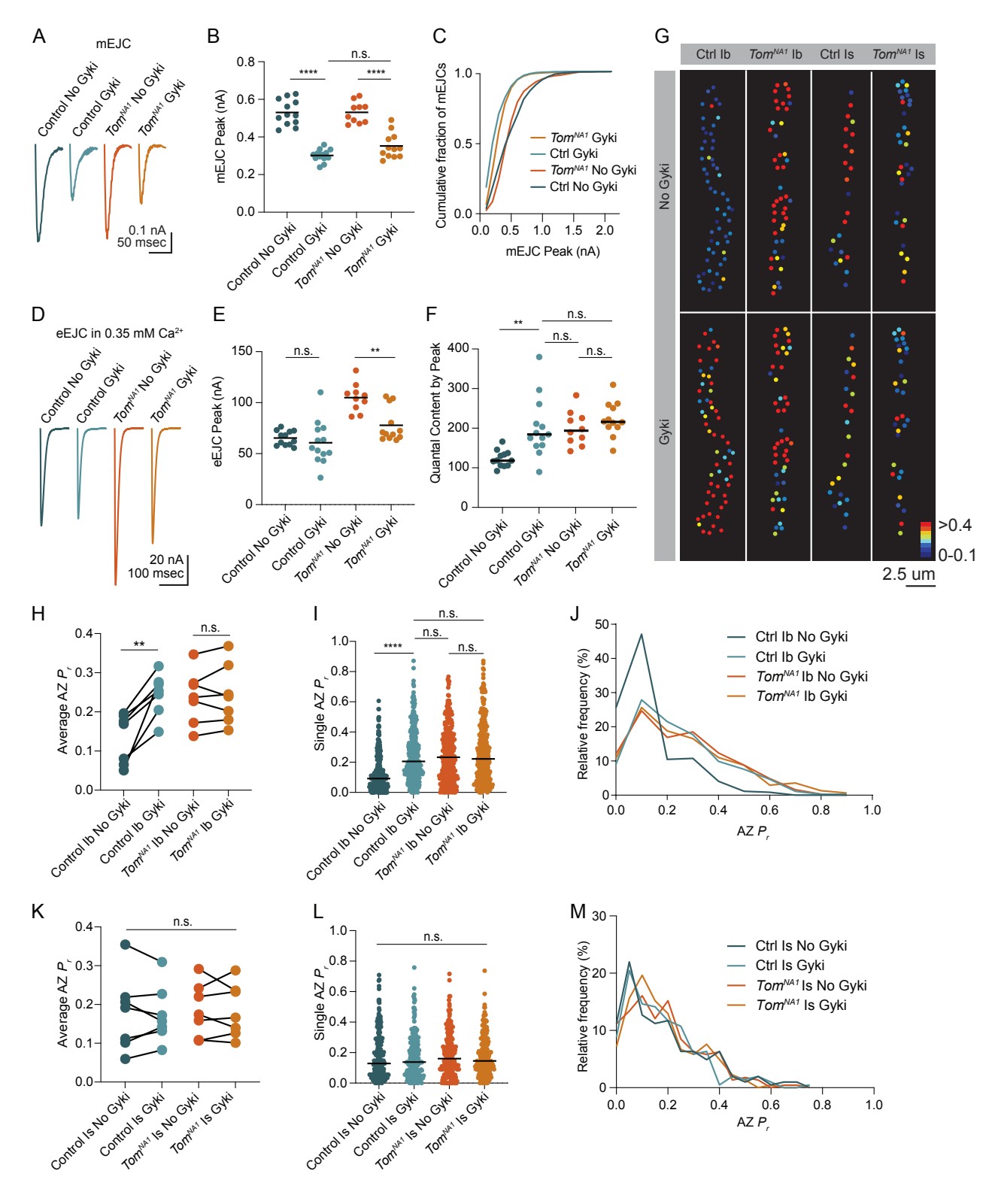

**Figure 8.** Tomosyn is essential for Gyki-induced presynaptic homeostatic potentiation (PHP). (**A**) Average mEJC amplitude in the presence and absence of the allosteric GluR inhibitor Gyki (10 uM). (**B**) Quantification of average mEJC peak current (nA) per NMJ (control, no Gyki: 0.5161, 0.5302 ± 0.01964 *n* = 12; control, Gyki: 0.3062, 0.3019 ± 0.009029, n = 13; *tomosyn*[NA1], no Gyki: 0.5318, 0.5312 ± 0.01789, n = 10; *tomosyn*[NA1], Gyki: 0.3373, 0.3521 ± 0.01934, n = 12; p < 0.0001; ≥ 7 larvae per group). (**C**) Histogram showing cumulative fraction of mEJCs by peak current. (**D**) Average eEJC peak amplitude (nA)

*Figure 8 continued on next page*

*Figure 8 continued*

following 15 min incubation in Gyki (10 uM). (**E**) Quantification of average eEJC peak (nA) per NMJ in 0.35 mM $Ca^{2+}$ (control, no Gyki: 65.02, 65.28 ± 2.062 *n* = 12; control, Gyki: 55.52, 60.67 ± 5.819, n = 13; *tomosyn*$^{NA1}$, no Gyki: 105.5, 104.9 ± 4.315, n = 10; *tomosyn*$^{NA1}$, Gyki: 70.10, 77.82 ± 4.778, n = 12; p < 0.0001; ≥ 7 larvae per group). (**F**) Quantification of average evoked quantal content per NMJ in 0.35 mM $Ca^{2+}$ approximated by peak current (control, no Gyki: 120.3, 124.9 ± 5.927 *n* = 12; control, Gyki: 185.7, 202.6 ± 20.54, n = 13; *tomosyn*$^{NA1}$, no Gyki: 195.2, 200.9 ± 13.47, n = 10; *tomosyn*$^{NA1}$, Gyki: 217.1, 224.2 ± 12.40, n = 12; p < 0.0001; ≥ 7 larvae per group). (**G**) Representative maps of AZ $P_r$ in Ib and Is before and after Gyki incubation following optical quantal imaging. (**H**) Average AZ $P_r$ per Ib NMJ before and after Gyki (control Ib, no Gyki: 0.1690, 0.1325 ± 0.02419, n = 7; control Ib, Gyki: 0.2538, 0.2451 ± 0.02049, n = 7; *tomosyn*$^{NA1}$ Ib, no Gyki: 0.2373, 0.2377 ± 0.02602, n = 7; *tomosyn*$^{NA1}$ Ib, Gyki: 0.2395, 0.2438 ± 0.02894, n = 7; p = 0.0094; ≥ 4 larvae per group). (**I**) Single AZ $P_r$ at Ib NMJs before and after Gyki (control Ib, no Gyki: 0.09200, 0.1275 ± 0.006387, n = 344; control Ib, Gyki: 0.2051, 0.2412 ± 0.009160, n = 344; *tomosyn*$^{NA1}$ Ib, no Gyki: 0.2325, 0.2515 ± 0.01016, n = 308; *tomosyn*$^{NA1}$ Ib, Gyki: 0.2220, 0.2601 ± 0.001117, n = 308; p < 0.0001; ≥ 4 larvae per group). (**J**) Histogram of single AZ $P_r$ at Ib NMJs before and after Gyki. (**K**) Average AZ $P_r$ per Is NMJ before and after Gyki (control Is, no Gyki: 0.1917, 0.1777 ± 0.03719, n = 7; control Is, Gyki: 0.1568, 0.1746 ± 0.02786, n = 7; *tomosyn*$^{NA1}$ Is, no Gyki: 0.1740, 0.1859 ± 0.02609, n = 7; *tomosyn*$^{NA1}$ Is, Gyki: 0.1662, 0.1843 ± 0.02598, n = 7; p = 0.9918; ≥ 4 larvae per group). (**L**) Single AZ $P_r$ at Is NMJs before and after Gyki (control Is, no Gyki: 0.1291, 0.1817 ± 0.01094, n = 205; control Is, Gyki: 0.1382, 0.1752 ± 0.009807, n = 205; *tomosyn*$^{NA1}$ Is, no Gyki: 0.1605, 0.1844 ± 0.009462, n = 224; *tomosyn*$^{NA1}$ Is, Gyki: 0.1454, 0.1813 ± 0.008662, n = 224; p = 0.9246; ≥ 4 larvae per group). (**M**) Histogram of single AZ $P_r$ at Is NMJs before and after Gyki. Complete data provided in *Figure 8—source data 1*.

The online version of this article includes the following figure supplement(s) for figure 8:

**Source data 1.** Source data for *Figure 8*.

**Figure supplement 1.** mEJC detection and non-saturation of quantal content following Gyki application.

**Figure supplement 1—source data 1.** Source data for *Figure 8—figure supplement 1*.

is predominantly expressed from Ib motoneurons. *Tomosyn* mutants displayed no significant increase in AZ $P_r$ from either Ib or Is terminals following Gyki application (*Figure 8G–M*). Together, these data indicate Gyki-induced PHP is Tomosyn-dependent and occurs exclusively at tonic Ib terminals. Loss of Tomosyn generates synaptic responses and a lack of PHP at tonic Ib NMJs that is similar to that observed in phasic Is neurons, indicating Tomosyn levels represent a key presynaptic mechanism for generating tonic versus phasic presynaptic output.

## Discussion

The findings reported here indicate the conserved presynaptic release suppressor Tomosyn functions in setting presynaptic output and plasticity differences for a tonic/phasic pair of motoneurons that co-innervate *Drosophila* larval muscles. CRISPR-generated mutations in *Drosophila tomosyn* revealed synchronous, asynchronous and spontaneous SV release are all elevated in the absence of the protein. While single evoked responses were enhanced, rapid depression of release was observed during train stimulation, suggesting loss of Tomosyn biases synapses toward a more phasic pattern of SV release. To directly test whether Tomosyn plays a unique role in tonic synapses, Ib and Is motoneurons were separately stimulated using optogenetics to measure their isolated contributions. These experiments revealed a 4-fold increase in output from Ib neurons with no change to Is release. Optical quantal analysis confirmed the Ib specific effect of Tomosyn and demonstrated enhanced evoked responses in *tomosyn* is due to higher intrinsic $P_r$ across the entire AZ population. Endogenously-tagged Tomosyn was more abundant at Ib synapses than Is, consistent with Tomosyn's role in regulating Ib release. Together, these data indicate the intrinsically high $P_r$ and rapid depression normally found in Is motoneurons is due in part to a lack of Tomosyn inhibition of SV usage at phasic synapses. High-frequency stimulation experiments demonstrate Tomosyn does not regulate the size of the immediately releasable SV pool (IRP) but rather regulates IRP usage to ensure sustained availability of SVs during prolonged stimulation, as the IRP is strongly biased towards early release in *tomosyn* mutants. We propose a model where *Drosophila* synapses are more phasic in release character by default, with tonic release requiring higher levels of Tomosyn to generate a fusion bottleneck that enables extended periods of stable release by slowing the rate of SV usage.

How Tomosyn normally suppresses SV release has been unclear (*Sakisaka et al., 2008*; *Yamamoto et al., 2010*; *Yizhar et al., 2007*; *Yizhar et al., 2004*). The most widely hypothesized mechanism is that Tomosyn competes with Syb2 for binding t-SNAREs. By forming fusion-incompetent SNARE complexes that must be disassembled by NSF, a pool of t-SNAREs is kept in reserve and can be mobilized by alleviating Tomosyn inhibition. Indeed, enhanced SNARE complex formation was found

in *Drosophila tomosyn* mutants, consistent with the model that Tomosyn's SNARE domain acts as a decoy SNARE to inhibit productive SNARE complex assembly. Expression of the Tomosyn scaffold alone failed to rescue the null phenotype, while overexpression of the scaffold had no effect on evoked release. As such, these data indicate that while the scaffold is required for full Tomosyn function, it does not directly inhibit fusion. Our observations are consistent with the mechanism proposed in *C. elegans*, but differ from studies in cultured mammalian cells suggesting the scaffold acts as an independent release suppressor by inhibiting Syt1 (*Burdina et al., 2011*; *Yamamoto et al., 2010*; *Yizhar et al., 2007*). Characterization of *Drosophila tomosyn/syt1* double mutants demonstrated Tomosyn suppresses release independent of Syt1, arguing the scaffold must serve a function that enhances the inhibitory activity of the SNARE domain independent of Syt1. Indeed, we found the Tomosyn SNARE motif was mislocalized without the WD40 scaffold, arguing this region indirectly supports Tomosyn's inhibitory activity by ensuring proper localization so the SNARE domain can compete for t-SNARE binding. Similar to studies in *C. elegans* and mammals, we find *Drosophila* Tomosyn co-localized with other SV proteins (*Geerts et al., 2017*; *McEwen et al., 2006*). Human Tomosyn transgenes also rescued elevated evoked and spontaneous release in *tomosyn* mutants, indicating functional conservation of its inhibitory properties. Overexpression of either *Drosophila* or human Tomosyn in a wild-type background also decreased release, demonstrating presynaptic output can be bi-directionally controlled by varying Tomosyn expression levels.

In addition to intrinsic release differences between tonic and phasic motoneurons, we found Tomosyn also controls presynaptic homeostatic potentiation (PHP). This form of synaptic plasticity occurs when presynaptic motoneurons upregulate $P_r$ and quantal content to compensate for decreased GluR function and smaller quantal size (*Böhme et al., 2019*; *Frank, 2014*; *Genç and Davis, 2019*; *Goel et al., 2019*; *Gratz et al., 2019*). Inducing PHP with the allosteric GluR inhibitor Gyki revealed Tomosyn is required for expression of this form of acute PHP at Ib terminals. Removing Tomosyn inhibition at Ib synapses generates a ~ 4 fold enhancement in evoked release, more than sufficient to compensate for a twofold reduction in evoked response size from two equally contributing motoneurons. Indeed, AZ $P_r$ mapping revealed Ib synapses potentiate in the presence of Gyki while Is terminals showed no change, indicating enhanced release from Ib is sufficient to homeostatically compensate for Gyki-induced decreases in quantal size. Although future studies will be required to determine the molecular cascade through which Tomosyn mediates PHP expression, prior work indicates PKA phosphorylation of Tomosyn reduces its SNARE binding properties and decreases its inhibition of SV release (*Baba et al., 2005*; *Ben-Simon et al., 2015*; *Chen et al., 2011*). Given Gyki-induced PHP expression requires presynaptic PKD (*Nair et al., 2020*), an attractive hypothesis is that PKD phosphorylates Tomosyn and reduces its ability to inhibit SNARE complex formation. Similar to *tomosyn* mutants, this could promote SV availability by generating a larger pool of free t-SNAREs to support enhanced docking of SVs at AZs. Increased docking would elevate single AZ $P_r$ by increasing the number of fusion-ready SVs upon $Ca^{2+}$ influx, similar to the effect we observed with quantal imaging.

Despite the importance of Tomosyn in regulating release character between tonic and phasic motoneurons, *tomosyn* null mutants are viable into adulthood. As such, the entire range of Tomosyn expression can be used by distinct neuronal populations *in vivo* to set presynaptic output. Tonic Ib terminals shift towards phasic release with no effect on Is output in *tomosyn* null mutants, resulting in a collapse of presynaptic release diversity between these two neuronal subgroups. Like *tomosyn*, null mutants in *syt7* are viable and show dramatically enhanced evoked release (*Fujii et al., 2021*; *Guan et al., 2020*). *Tomosyn/syt7* double mutants show even greater increases in release output, arguing multiple non-essential presynaptic proteins can independently fine tune synaptic strength within the presynaptic terminal. Together, these experiments demonstrate Tomosyn is a highly conserved release inhibitor that varies in expression between distinct neuronal subtypes to regulate intrinsic $P_r$ and plasticity, providing a robust mechanism to generate presynaptic diversity across the nervous system.

## Materials and methods
### *Drosophila* stocks

*Drosophila melanogaster* were cultured on standard medium between 22°C and 25°C. Third instar larvae were used for all *in vivo* and immunostaining experiments. Adult brain extracts were used for western blot analysis. Males were preferentially used in this study to facilitate genetic crossing

schemes and avoid sex-specific phenotypic differences. *Tomosyn* null mutants used in the study include *tomosyn^NA1* (this study)**,** *tomosyn^FS1* (this study), and Df(1)ED7161 (Bloomington *Drosophila* Stock Center (BDSC) #9217). Strains used for rescue experiments include *elav^C155*-GAL4 (BDSC#8765), UAS-Tom13A-6xMyc (this study), UAS-Tom13A-ΔSNARE-6xMyc (this study), UAS-Tom13B-6xMyc (this study), UAS-Tom13B-ΔSNARE-6xMyc (this study), UAS-4xMyc-TomSNARE (this study), and UAS-HumanTom1-6xMyc (this study). Double mutant experiments were performed with *syt1^AD4* (*DiAntonio and Schwarz, 1994*), *syt1^N13* (*Littleton et al., 1993*), *syt7* control (*Guan et al., 2020*), and *syt7^M1* (*Guan et al., 2020*). For single neuron optical stimulation experiments, the Ib-specific Gal4 driver GMR94G06 (BDSC #40701) and the Is-specific Gal4 driver GMR27F01 (BDSC #49227) was used to drive expression of UAS-ChR2-T159C (*Dawydow et al., 2014*) in Ib or Is motoneurons innervating larval muscle 1. For AZ $P_r$ mapping experiments, Mef2-Gal4 (BDSC #27390), 44H10-LexA (provided by Gerry Rubin), LexAOp-myr-jGCaMP7S (this study), UAS-myr-jGCaMP8s (this study), GluRIIA-RFP (provided by Stephan Sigrist), and GluRIIB-GFP (provided by Stephan Sigrist) transgenic lines were used.

## Genome engineering and UAS/LexA constructs

To generate *tomosyn^NA1*, two guide RNAs (gRNAs) flanking the *tomosyn* locus were selected using the CRISPR Optimal Target Finder (*Gratz et al., 2014*). These gRNAs were fused with the pCFD4 expression vector (Addgene #49411) (*Port et al., 2014*) according to the Gibson assembly protocol using NEBuilder HighFidelity DNA Assembly Cloning Kit (E5520). Gibson assembly was used to generate a donor construct encoding a floxed P3> DsRed reporter cassette (Addgene #51434) flanked with homology arms directly outside of the *tomosyn* gene isolated using PCR. These constructs were co-injected into vasa-Cas9 embryos (BDSC #56552) and DsRed positive transformants were selected by BestGene Inc (Chino Hills, CA, USA). To generate *tomosyn^FS1*, the pCFD4 gRNA construct was injected without a donor, and frame shift mutants were identified by PCR and sequencing. The Cas9 chromosome was removed from both lines by backcrossing to w^-/- (BDSC #3605). For both *tomosyn^NA1* and *tomosyn^FS1*, unmodified progeny of the CRISPR-injected embryos were used as genetic background controls. To generate *tomosyn^13A-Clover*, gRNAs targeting exon13A of *tomosyn* were cloned into pCFD5 (Addgene #73914) (*Port and Bullock, 2016*) and co-injected with a donor plasmid by BestGene Inc The donor was made by amplifying homology arms from the genome by PCR and fusing them by Gibson assembly with a cDNA coding for 6xHis-mClover3 (Addgene #74252) (*Bajar et al., 2016*) in frame with exon 13 A. To generate rescue constructs, the relevant cDNAs were synthesized by GENEWIZ, Inc (South Plainfield, NJ, USA) and cloned into pBid-UASc (Addgene #35200) (*Wang et al., 2012*) using EcoRI and XbaI. These constructs were injected into embryos containing the VK27 attP acceptor site by BestGene, Inc (BDSC #9744). Positive transformants were selected and balanced. The fluorescent $Ca^{2+}$ sensor GCaMP7s was tethered to the plasma membrane with an N-terminal myristoylation (myr) sequence. A cDNA encoding the first 90 amino acids of Src64b, containing a myristoylation target sequence, was PCR amplified from the pBid-UAS-myr plasmid (*Akbergenova et al., 2018*) and fused with the GCaMP7s cDNA (Addgene # 104463) (*Dana et al., 2019*) and EcoRI/XbaI digested pBid-LexA (a gift from Brian McCabe) using Gibson assembly. pBid-UAS-myr-jGCaMP8s was made by fusing a GCaMP8s cDNA (Addgene # 162374) (*Zhang et al., 2020*) with BglII/XbaI digested pBid-UAS-myr using Gibson assembly. These constructs were injected by BestGene, Inc into embryos containing the attP2 acceptor site and positive transformants were isolated (BDSC #8622).

## Bioinformatics

NCBI BLAST was used to identify homologs of *Drosophila* nSyb and Tomosyn in *C. elegans*, *N. vectensis*, *M. lignano*, *O. sinensis*, *C. teleta*, *A. planci*, *D. rerio*, *M. musculus*, and *H. sapiens*. The C-terminal tail of *S. cerevisiae* Sro7 was used as the outgroup. UCSC Genome Browser's Cons 124 feature was used to assess sequence conservation with *Drosophila tomosyn* as the reference sequence (htpps://genome.ucsc.edu/). The Póle Rhône-Alpes de Bioinformatique (PRABI; https://npsa-prabi. ibcp.fr) coiled-coil prediction tool was used to identify the C-terminal SNARE domain of each protein and the BLOSUM62 algorithm of the Matlab 2020a seqpdist function was used to create sequence alignment. Phylogenetic trees were generated with the seqlinkage Matlab function.

Protein sequences used for alignment and phylogenetic tree construction:

| Protein | Species | NCBI accession number |
|---|---|---|
| Tomosyn | *N. vectensis* | EDO30312.1 |
| | *M. lignano* | PAA82513.1 |
| | *O. sinensis* | XP_036368981.1 |
| | *C. teleta* | ELU03639.1 |
| | *A. planci* | XP_022100438.1 |
| | *D. rerio* | XP_021334414.1 |
| | *M. musculus* | XP_006512991.1 |
| | *H. sapiens* | NP_001121187.1 |
| | *D. melanogaster* | NP_001162735.1 |
| | *C. elegans* | AAX89146.1 |
| Synaptobrevin/ VAMP2 | *N. vectensis* | XP_001634446.2 |
| | *M. lignano* | PAA92592.1 |
| | *O. sinensis* | XP_029648798.1 |
| | *C. teleta* | ELU12629.1 |
| | *A. planci* | XP_022085538.1 |
| | *D. rerio* | NP_956299.1 |
| | *M. musculus* | NP_033523.1 |
| | *H. sapiens* | NP_001317054.1 |
| | *D. melanogaster* | NP_477058.1 |
| | *C. elegans* | NP_001379956.1 |
| | *S. cerevisiae* | NP_594120.1 |
| Sro7 | *S. cerevisiae* | NP_015357.1 |

## Western blot analysis and immunocytochemistry

Western blotting of adult head lysates (ten heads per/lane) was performed using standard laboratory procedures with mouse anti-Syx1a (8C3, 1:1000; Developmental Studies Hybridoma Bank (DSHB, Iowa City, IA)) anti-Myc (GeneTex: GTX29106, 1:1000) and mouse anti-Tub (Sigma: T5168, 1:1,000,000). The boiling step was omitted to preserve the 7 S complex. IR Dye 680LT-conjugated goat anti-mouse (1:10,000, LICOR; 926–68020) was used as the secondary antibody. Visualization was performed with a LI-COR Odyssey Imaging System (LI-COR Biosciences, Lincoln, MA, USA) and analysis was performed using the Plot Lanes and Measure Areas function of FIJI image analysis software (*Schindelin et al., 2012*). Lanes with poor protein loading were excluded from analysis as described in the source data and statistics supplementary file.

Immunostaining for AZ and bouton counting was performed on wandering 3rd instar larvae dissected in $Ca^{2+}$-free HL3.1 and fixed for 7 min in $Ca^{2+}$-free HL3.1 containing 4 % PFA. Larvae were blocked and permeabilized for 1 hr in PBS containing 0.1 % Triton X-100, 2.5 % NGS, 2.5 % BSA and 0.1 % sodium azide. Larvae were incubated overnight with primary antibody at 4 °C and 2 hr in secondary antibody at room temperature. Samples were mounted on slides with Vectashield (Vector Laboratories, Burlingame, CA). Antibodies used for immunolabeling were: rabbit anti-GFP at 1:1000 (ab6556; Abcam, Cambridge, UK), mouse anti-BRP at 1:500 (Nc82; DSHB), mouse anti-Synapsin at 1:500 (3C11; DSHB), rabbit anti-Myc at 1:500 (GTX29106; GeneTex, Irvine, CA, USA), rabbit anti-Syt1 (gift of Noreen Reist) at 1:500, and DyLight 649 conjugated anti-HRP at 1:1000 (#123-605-021; Jackson Immuno Research, West Grove, PA, USA). Secondary antibodies for morphology and co-localization experiments were used at 1:500: goat anti-rabbit Alexa Fluor 488-conjugated antibody (A-11008; Thermofisher) and goat anti-mouse Alexa Fluor 546-conjugated antibody (A-11030; ThermoFisher).

The secondary antibody used for anti-GFP staining was goat anti-rabbit Alexa Flour 488-conjugated antibody (A-11008; Thermofisher) used at 1:500. Immunoreactive proteins were imaged at segments A3 and A4 of muscle fiber four for all experiments, except for anti-GFP staining, which was imaged at muscles 6/7. Images were acquired on a PerkinElmer Ultraview Vox spinning disk confocal microscope system using a 63 x oil immersion objective. Ib and Is terminals were identified based on bouton and NMJ size, with Is having characteristically smaller boutons and total NMJ size. NMJ morphology, staining intensity, and co-localization between channels were analyzed using Volocity 6.3.1 software.

## Electrophysiology

Postsynaptic currents from the indicated genotypes were recorded from 3rd instar larvae at muscle fiber 6 (unless otherwise noted) of segment A4 using two-electrode voltage clamp with a −80 mV holding potential in HL3.1 saline solution (in mM, 70 NaCl, 5 KCl, 10 NaHCO3, 4 MgCl2, 5 trehalose, 115 sucrose, 5 HEPES, pH 7.18) as previously described (*Jorquera et al., 2012*). Final $[Ca^{2+}]$ was adjusted to the level indicated. All electrophysiology experiments were performed at room temperature. Inward currents recorded during TEVC are labeled on a reverse axis in the figures for simplicity. Asynchronous release contribution was approximated by fitting the weighted average of two logarithmic regressions with separate time constants to the normalized cumulative charge transfer of evoked responses as previously described (*Jorquera et al., 2012*). The $Ca^{2+}$ cooperativity of release was determined from the Hill coefficient of a 4-parameter logistic regression of evoked responses fit to the linear range (0.1–0.75 mM $Ca^{2+}$). Data acquisition and analysis was performed using Axoscope 9.0 and Clampfit 9.0 software (Molecular Devices, Sunnyvale, CA, USA). mEJCs were analyzed with Mini Analysis software 6.0.3 (Synaptosoft, Decatur, GA, USA). Motor nerves innervating the musculature were severed and placed into a suction electrode. Action potential stimulation was applied at 0.33 Hz (unless indicated) using a programmable stimulator (Master8, AMPI; Jerusalem, Israel).

Optogenetic experiments were performed in the same way with the following modifications. Postsynaptic currents were recorded from 3rd instar larvae at segment A4 of muscle fiber 1. Evoked postsynaptic currents were generated using the Master8 stimulator and an LED driver (LED-D1B, THORLABS, Newton, NJ, USA) to generate 470 nm light pulses from an attached LED (M470F3, THORLABS, Newton, NJ, USA). Ib and Is currents were separately evoked by driving expression of ChR2 (UAS-ChR2-T159C, provided by Robert Kittel) under the control of GMR94G06-Gal4 (BDSC #40701) or GMR27F01-Gal4 (BDSC# 49227), respectively.

## Gyki application and PHP analysis

Gyki was diluted fresh each day in HL3.1 to a final concentration of 10 µM. The final $Ca^{2+}$ concentration was adjusted to the level indicated. The Gyki solution was bath applied to fully dissected larvae for 15 minutes as previously described (*Nair et al., 2020*). Subsequent recordings were performed in the continued presence of bath applied Gyki. Gyki was used instead of the GluR blocker Philanthotoxin-433 (PhTX), as PhTX requires a partially dissected preparation capable of muscle contraction for PHP induction. This more intact preparation is not compatible with imaging AZ release before and after PHP. In contrast, PHP expression can occur in a fully stretched preparation following Gyki application.

## Optical AZ *Pr* mapping

AZ $P_r$ mapping experiments were performed on a Zeiss Axio Imager equipped with a spinning-disk confocal head (CSU-X1; Yokagawa, Japan) and ImagEM X2 EM-CCD camera (Hamamatsu, Hamamatsu City, Japan) as previously described (*Akbergenova et al., 2018*). For $P_r$ mapping of *tomosyn*[FS1], myristoylated-GCaMP7s was expressed in larval muscles with GMR44H10-LexA (provided by Gerald Rubin). Individual PSDs were visualized at segments A2-A4 of muscle fiber four by expression of GluRIIA-RFP and GluRIIB-GFP (hereafter referred to as GluR) under control of their endogenous promoters (provided by Stephan Sigrist). An Olympus LUMFL N 60 X objective with a 1.10 NA was used to acquire GCaMP7s imaging data at 8 Hz. Third instar larvae were dissected in $Ca^{2+}$-free HL3 containing 20 mM MgCl₂. After dissection, preparations were maintained in HL3 with 20 mM MgCl₂ and 1.0 mM $Ca^{2+}$ for 5 min. A dual channel multiplane stack was imaged at the beginning of each experiment to identify GluR-positive PSDs. Single focal plane videos were then recorded while motoneurons innervating the muscles were stimulated with a suction electrode at 0.3 Hz for 3 min. GluR PSD position was

re-imaged every 25 s during experimentation. The dual channel stack was merged with single plane images using the max intensity projection algorithm from Volocity 6.3.1 software. The position of all GluR PSDs was then added to the myr-GCaMP7s stimulation video. GluR positive PSDs were detected automatically using the spot finding function of Volocity and equal size ROIs were assigned to the PSD population. In cases where the software failed to label visible GluR PSDs, ROIs were added manually. GCaMP7s peak flashes were detected and assigned to ROIs based on centroid proximity. Evoked events were identified as frames with three or more simultaneous GCaMP events across the arbor. The time and location of $Ca^{2+}$ events were imported into Excel or Matlab for further analysis. Evoked GCaMP events per ROI were divided by the number of stimulations to calculate AZ $P_r$.

AZ $P_r$ experiments with Gyki were performed in the same way with the following modifications. Mef2-Gal4 (BDSC #27390) was used to drive expression of UAS-myr-GCaMP8s in larval muscles. Dissected preparations were maintained in HL3 containing 10 mM $MgCl_2$ and 0.5 mM $Ca^{2+}$ and imaged at muscle fibers 6/7. The HL3 solution was exchanged for an identical solution containing 10 µM Gyki and incubated for 15 min. A second imaging session was recorded at each NMJ after Gyki incubation. AZ locations were identified by labeling peaks for all events and regions of highest peak densities were assigned as ROIs. Release events were assigned to ROIs using the centroid proximity algorithm in Volocity 6.3.1.

## Electron microscopy

*Tomosyn^NA1* and control 3rd instar larvae were dissected in $Ca^{2+}$-free HL3 saline and fixed in 1 % glutaraldehyde, 4 % formaldehyde and 0.1 M sodium cacodylate buffered saline (CBS) with 1 mM magnesium chloride for 10 min at room temperature as previously described (*Akbergenova and Bykhovskaia, 2009*). After fixative exchange, samples were microwaved in a BioWave Pro Pelco (Ted Pella, Inc, Redding, CA, USA) using the following fixation protocol: (1) 100 W 1 min, (2) 1 min off, (3) 100 W 1 min, (4) 300 W 20 s, (5) 20 s off, (6) 300 W 20 s. Steps 4–6 were repeated twice more. Samples were then washed in CBS and stained *en bloc* for 30 min in 1 % osmium tetroxide. Following another CBS wash, samples were stained *en bloc* for 30 min in 2 % uranyl acetate and briefly incubated in sequentially anhydrous solutions of ethanol and then pure anhydrous acetone. Epoxy resin infiltration was performed by incubating the dehydrated samples in a series of acetone/epoxy mixtures, with the acetone percentage decreasing in each successive step (Embed 812; Electron Microscopy Sciences). Thin sections (40–50 nm) were collected on Formvar/carbon-coated copper slot grids and stained on grid for ~5 min with lead citrate. Sections were imaged at ×49,000 magnification at 120 kV using a Tecnai G2 electron microscope (FEI, Hillsboro, OR, USA) equipped with a charge-coupled device camera (Gatan, Pleasanton, CA, USA). Micrographs of type Ib boutons from segment 4 of muscle fibers 6/7 were analyzed using Volocity 6.3.1. SV centers were annotated as points, T-bar bases as single pixel ROIs, and electron densities as contoured lines. Distances between these features were calculated using the Measure Distance function to determine SV spacing, SV number, and docked SV number (SVs with centers<50 nm to the electron dense AZ).

## Quantification and statistical analysis

Statistical analysis and graphing were performed with GraphPad Prism (San Diego, CA, USA). In two cases, outliers were identified and removed using the default settings of the Identify Outlier function in Prism9 (mini frequency of elav-Gal4,*tomosyn^NA1* in *Figure 2Q*, excluded mini frequency was 23.3 Hz; mini frequency of *syt7^M1* in *Figure 4L*, excluded mini frequency was 6.20 Hz). Electrophysiological traces were generated using the plot function in Matlab R2020A (MathWorks, Natick, MA, USA). Statistical significance was determined using Student's *t* test for comparisons between two groups, or a One-way ANOVA followed by Tukey's multiple comparisons test for comparisons between three or more groups unless noted. In the figures, the center of each distribution is plotted as the median value and reported in the figure legends as the median, mean ± SEM, *n*. In the main text, the centers and *n* are reported as mean ± SEM, *n*. In all cases, *n* represents the number of individual NMJs analyzed unless otherwise noted. The number of larvae used per group in each experiment is indicated in the figure legends. Asterisks in the figures denote p-values of: *, $p \leq 0.05$; **, $p \leq 0.01$; ***, $p \leq 0.001$; and ****, $p \leq 0.0001$. The Source Data and Statistical Analysis excel file contains individual spreadsheets labeled with figure number and includes all primary source data and statistical comparisons.

## Acknowledgements

We thank the Bloomington *Drosophila* Stock Center (NIH P40OD018537), Stephan Sigrist (Freie Univesitat Berlin), Brian McCabe (Brain Mind Institute, EPFL-Swiss Federal Institute of Technology) and Gerry Rubin (Janelia Research Campus) for providing *Drosophila* stocks, Dina Volfson (MIT) for transgenic stock generation, and members of the Littleton lab for helpful discussion and comments. This work was supported by NIHs grants NS40296 and MH104536 to JTL. KLC, and NAS were supported in part by NIH pre-doctoral training grant T32GM007287.

## Additional information

### Funding

| Funder | Grant reference number | Author |
| --- | --- | --- |
| National Institute of Neurological Disorders and Stroke | NS040296 | J Troy Littleton |
| National Institute of Mental Health | MH104536 | J Troy Littleton |
| National Institutes of Health | T32GM007287 | Karen L Cunningham<br>Nicole A Aponte-Santiago |

The funders had no role in study design, data collection and interpretation, or the decision to submit the work for publication.

### Author contributions

Chad W Sauvola, Conceptualization, Data curation, Formal analysis, Investigation, Methodology, Writing - original draft, Writing - review and editing; Yulia Akbergenova, Data curation, Formal analysis, Methodology, Writing - review and editing; Karen L Cunningham, Data curation, Formal analysis, Writing - review and editing; Nicole A Aponte-Santiago, Resources, Writing - review and editing; J Troy Littleton, Conceptualization, Formal analysis, Funding acquisition, Project administration, Supervision, Writing - original draft, Writing - review and editing

### Author ORCIDs

J Troy Littleton http://orcid.org/0000-0001-5576-2887

### Decision letter and Author response

Decision letter https://doi.org/10.7554/eLife.72841.sa1
Author response https://doi.org/10.7554/eLife.72841.sa2

## Additional files

### Supplementary files

• Transparent reporting form

### Data availability

All data generated or analysed during this study are included in the manuscript and supporting figures.

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
