## [Editor Report]

Sauvola and colleagues define the function of Tomosyn in establishing release probability at NMJ synapses in *Drosophila* larval NMJs. They present compelling evidence that loss of Tomosyn results in increased evoked and spontaneous release. They further find that Tomosyn likely acts as a decoy SNARE protein independently of Syt 1 and Syt 7 to negatively regulate SV docking. The data are of no doubt interesting for researchers in the synaptic transmission field.

---

## [Decision Letter]

**Decision letter after peer review:**

Thank you for submitting your article "The decoy SNARE Tomosyn sets tonic versus phasic release properties and is required for homeostatic synaptic plasticity" for consideration by *eLife*. Your article has been reviewed by 2 peer reviewers, and the evaluation has been overseen by a Reviewing Editor and Claude Desplan as the Senior Editor. The following individual involved in review of your submission has agreed to reveal their identity: Heather Broihier (Reviewer #1).

The reviewers have discussed their reviews with one another, and the Reviewing Editor has drafted this to help you prepare a revised submission. Based on the suggestions of the reviewers it would be appropriate to address their concerns with textual changes.

*Reviewer #1:*

In this manuscript, Sauvola and colleagues define the function of Tomosyn in establishing release probability at NMJ synapses in the larval *Drosophila* model. They present compelling evidence that loss of Tomosyn results in increased evoked and spontaneous release. They further find that Tomosyn likely acts as a decoy SNARE protein independently of Syt 1 and Syt 7 to negatively regulate SV docking. They find that Tomosyn is selectively expressed in 1b motorneurons where it is key to setting the tonic release properties of this motorneuron subtype.

This manuscript was a delight to read! The data are exciting and logically presented. Of particular note, the figures are beautiful and exceptionally well done. The experiments are rigorously executed and appropriately controlled. These findings will be of significant interest to neurobiologists interested in mechanisms of NT release as well as to those interested in how distinct populations of neurons exhibit distinct and characteristic release properties. These are fundamental questions in the field and are also likely to have implications for neurological diseases.

1. In the text accompanying Figure 3, the authors indicate that independent expression of the scaffold and SNARE domains does not reconstitute full-length Tomosyn function, indicated the domains need to be physically connected. This is an interesting result, but I do not actually see the corresponding genetic background (elavGAL4, UAS-SNARE + UASDeltaSNARE) indicated in Figure 3A.

2. The differential localization of Tom in 1b versus 1s boutons is striking and fits very neatly with the functional data. It would seem important to demonstrate that the Clover tag did not disrupt Tom function by recording evoked and spontaneous activity from knockin homozygotes.

3. Could the authors indicate why they chose to use the GluR inhibitor Gyki to induce PHP in lieu of the more commonly used blocker PhTX? The authors are leaning on a 2020 preprint from the Muller lab (Nair et al., 2020) to justify the use of this drug, but this study is apparently not yet through peer-review, raising some questions about the specificity of Gyki.

4. Did the authors observe any differences in post-synaptic differentiation in Tom mutants? The central finding of the study that Tom expression helps define the physiological attributes of the 1b subtype made this reviewer wonder if its effects were confined to the presynaptic compartment, or whether it might also indirectly regulate the stereotyped difference in Dlg expression levels that mark 1b terminals.

*Reviewer #2:*

Sauvola et al. present a thorough analysis of tomosyn function at the *Drosophila* neuromuscular junction. Tomosyn is a SNARE-interacting protein that has previously been shown to reduce release probability in vertebrate and *C. elegans* model systems, as well as to some extend using knockdown, in flies. The present study represents the first in-depth mutant analysis in flies, including a structure-function analysis, detailed electrophysiology, biochemistry, genetics (double mutants), EM and optogenetic analyses. Most findings confirm and deepen previous results from other systems. In addition, by distinguishing tonic and phasic boutons at the larval NMJ relate the confirmed role of Tomosyn as a suppressor of release probability to the tonic and phasic modes of firing (more Tomosyn=reduced release probability at tonic synapses; less Tomosyn=increased release probability at phasic synapses). Overall the study presents no real surprises, but represents the most detailed study and certainly first clear fly reference for Tomosyn function.

Tomosyn has previously been studied in the vertebrate and *C. elegans* system at some depth. *C. elegans* tomosyn mutants had been shown to lead to more docked vesicles (Gracheva et al., Neurosci Lett. 2006) and prolonged release sue to increase SV exocytosis (Hu et al., *eLife* 2013 (not cited)). In vertebrate neurons, Tomosyn is enriched at synapses with low release probability and has been shown to regulate release probability optogenetically (Ben-Simon et al., Cell Rep. 2015) and mutants increase release efficacy from the readily releasable pool (Cazares et al., J Neurosci. 2016 (not cited)). In *Drosophila*, knock-down of tomosyn has been shown to increase synaptic strength at the neuromuscular junction (Chen et al., PNAS 2011). In light of the published literature, the new findings by Sauvola et al., are confirmatory, but also add substantially more detailed analyses as well as the perspective of differential tomosyn functions at tonic (type 1b) and phasic (type 1s) motoneurons. The structure function analysis provide new insights into how Tomosyn suppresses SV release.

There is a lot in the manuscript that should serve as a reference for tomosyn function in flies. I found both experiments and their discussion convincing, only slightly overemphasizing the novelty associated with the findings. I have no major critique of this paper. However, I have two suggestions that could further improve the manuscript:

1. The idea that levels of Tomosyn and direct modulators of release probability at type 1s and 1b boutons would be shown best by directly relating levels of endogenous Tomosyn following different modulated overexpression experiments with electrophysiological properties. How linearly do increased levels of tomosyn (ideally comapred to levels of SNAREs) lead to exocytic suppression?

2. In a similar vein: the authors had previously established single synapse optic measurements of SV release at the NMJ. It would seem to me an almost ideal experiment to do this in conjunction with endogenously tagged tomosyn at the level of individual type 1s and 1b boutons to show the local, direct and quantitative relationship of individual release events with tomosyn levels.

---

## [Author Response]

Reviewer #1:[…] 1. In the text accompanying Figure 3, the authors indicate that independent expression of the scaffold and SNARE domains does not reconstitute full-length Tomosyn function, indicated the domains need to be physically connected. This is an interesting result, but I do not actually see the corresponding genetic background (elavGAL4, UAS-SNARE + UASDeltaSNARE) indicated in Figure 3A.

We thank the reviewer for catching this omission during figure preparation. We have corrected the axis label in revised Figure 3A.

2. The differential localization of Tom in 1b versus 1s boutons is striking and fits very neatly with the functional data. It would seem important to demonstrate that the Clover tag did not disrupt Tom function by recording evoked and spontaneous activity from knockin homozygotes.

We designed the Clover tag to insert into a predicted unstructured region in one of the two major *tomosyn* isoforms (*tomosyn 13A*) to minimize any disruption in function. Since we were unable to isolate a knockin for the other *tomosyn* isoform (*tomosyn 13B*), we would not be able to unambiguously conclude that the Clover tag doesn't disrupt Tomosyn function, given Tomosyn 13B would be unaffected and can rescue release on its own. We did not use the tagged line for any functional data requiring Tomosyn activity, only as a marker for localization. We believe the functional data presented in Figure 7H-I provides strong evidence that the difference in expression level observed for *tomosyn^13A-Clover^* between 1b and 1s reflects a biologically relevant difference in its function between these motoneuron subtypes.

3. Could the authors indicate why they chose to use the GluR inhibitor Gyki to induce PHP in lieu of the more commonly used blocker PhTX? The authors are leaning on a 2020 preprint from the Muller lab (Nair et al., 2020) to justify the use of this drug, but this study is apparently not yet through peer-review, raising some questions about the specificity of Gyki.

Gyki was used instead of PhTX as we needed to monitor the expression of PHP before and after drug application at the level of individual AZs (Figure 8G-M). PHP expression can occur in a fully stretched preparation following Gyki application, allowing imaging of single AZ release probability before and after plasticity induction. In contrast, PhTX is a much more difficult reagent to use and requires a partially dissected relaxed preparation capable of muscle contraction for PHP expression to occur. This preparation is not compatible with imaging SV fusion events before and after application at the same AZ. The requirement to align release probability maps to single AZs at the same NMJ made Gyki the only choice for release mapping experiments in Figure 8G. We have added this information to the methods section so readers know why we chose Gyki over PhTX.

4. Did the authors observe any differences in post-synaptic differentiation in Tom mutants? The central finding of the study that Tom expression helps define the physiological attributes of the 1b subtype made this reviewer wonder if its effects were confined to the presynaptic compartment, or whether it might also indirectly regulate the stereotyped difference in Dlg expression levels that mark 1b terminals.

Indeed, some reports in mammals suggest Tomosyn may have a secondary role in regulating vesicle trafficking within the post-synaptic compartment. By RNA profiling in *Drosophila* larvae, we find that Tomosyn is expressed at far higher levels in neurons versus muscles (>20-fold). We also did not observe postsynaptic expression with our CRISPR-tagged construct. Likewise, the phenotypes of enhanced evoked and spontaneous release are rescued by presynaptic expression of Tomosyn with *elav*-GAL4 (Figure 2P,Q), so we don’t have any data to support a postsynaptic role at the fly NMJ. However, this will be an interesting area for future study of postsynaptic cargo trafficking, and is not something we addressed in the current study.

Reviewer #2:[…] There is a lot in the manuscript that should serve as a reference for tomosyn function in flies. I found both experiments and their discussion convincing, only slightly overemphasizing the novelty associated with the findings. I have no major critique of this paper. However, I have two suggestions that could further improve the manuscript:1. The idea that levels of Tomosyn and direct modulators of release probability at type 1s and 1b boutons would be shown best by directly relating levels of endogenous Tomosyn following different modulated overexpression experiments with electrophysiological properties. How linearly do increased levels of tomosyn (ideally comapred to levels of SNAREs) lead to exocytic suppression?

We agree this is an interesting question. The data presented in Figure 3 Supplement 1 and in Figure 2P-S partially address this question. In Figure 3 Supplement 1, we show that overexpression of Tomosyn using a pan-neuronal Gal4 driver reduces both evoked and spontaneous release below control levels, suggesting higher levels of Tomosyn more effectively suppress release. In Figure 2P, we show both *Drosophila* and human Tomosyn rescue the null phenotype. However, the human Tomosyn construct appears to over-rescue by further reducing release, consistent with the greater level of protein expression of human Tomosyn at the synapse (Figure 2S). We also performed preliminary experiments to determine if artificial overexpression of Tomosyn in 1s motoneurons using Gal4/UAS could trigger more tonic-like release from these phasic terminals. However, we were unable to drive Tomosyn expression in Is terminals following overexpression, suggesting post-translational mechanisms are acting to degrade the protein and prevent Tomosyn expression in this cell type. Future work targeting potential degradation mechanisms would be very exciting and could reveal additional mechanisms for presynaptic plasticity beyond transcriptional control.

2. In a similar vein: the authors had previously established single synapse optic measurements of SV release at the NMJ. It would seem to me an almost ideal experiment to do this in conjunction with endogenously tagged tomosyn at the level of individual type 1s and 1b boutons to show the local, direct and quantitative relationship of individual release events with tomosyn levels.

Although very interesting, this experiment is not feasible with our current toolkits. The endogenously Clover-tagged Tomosyn is not bright enough to observe live and requires counterstaining. As such, this prevents examination of dynamic Tomosyn levels at individual release sites. In addition, immunochemistry on fixed preparations as shown in Figure 3C reveals a Tomosyn distribution that is relatively homogenous across individual release sites at this level of resolution. Given the uniform increase in *Pr* across the entire AZ population in *tomosyn* null mutants (Figure 7K), we would not expect Tomosyn to act as a key regulator of heterogeneity at the level of individual AZs.